# Brown adipose tissue is the key depot for glucose clearance in microbiota depleted mice

Min Li[1,2,3,9], Li Li [1,2,4,9], Baoguo Li[1,2,5,9], Catherine Hambly[3], Guanlin Wang[1,2,3], Yingga Wu[1,2,3], Zengguang Jin[1], Anyongqi Wang[1,2], Chaoqun Niu[1], Christian Wolfrum [6] & John R. Speakman [1,3,7,8 ✉]

Gut microbiota deficient mice demonstrate accelerated glucose clearance. However, which tissues are responsible for the upregulated glucose uptake remains unresolved, with different studies suggesting that browning of white adipose tissue, or modulated hepatic gluconeogenesis, may be related to enhanced glucose clearance when the gut microbiota is absent. Here, we investigate glucose uptake in 22 different tissues in 3 different mouse models. We find that gut microbiota depletion via treatment with antibiotic cocktails (ABX) promotes glucose uptake in brown adipose tissue (BAT) and cecum. Nevertheless, the adaptive thermogenesis and the expression of uncoupling protein 1 (UCP1) are dispensable for the increased glucose uptake and clearance. Deletion of Ucp1 expressing cells blunts the improvement of glucose clearance in ABX-treated mice. Our results indicate that BAT and cecum, but not white adipose tissue (WAT) or liver, contribute to the glucose uptake in the gut microbiota depleted mouse model and this response is dissociated from adaptive thermogenesis.

[1] State Key Laboratory of Molecular Developmental Biology, Institute of Genetics and Developmental Biology, Chinese Academy of Sciences, Beijing, PR China. [2] University of Chinese Academy of Sciences, Beijing, PR China. [3] Institute of Biological and Environmental Sciences, University of Aberdeen, Aberdeen, Scotland, UK. [4] Hypothalamic Research Center, Department of Internal Medicine, UT Southwestern Medical Center, Dallas, TX, USA. [5] Department of Immunology, Weizmann Institute of Science, Rehovot, Israel. [6] Institute of Food Nutrition and Health and Department of Health Sciences and Technology (ETH), Schwerzenbach, Switzerland. [7] CAS Center for Excellence in Animal Evolution and Genetics (CCEAEG), Beijing, PR China. [8] Shenzhen Key Laboratory of Metabolic Health, Center for Energy Metabolism and Reproduction, Shenzhen Institutes of Advanced Technology, Chinese Academy of Sciences, Shenzhen, PR China. [9] These authors contributed equally: Min Li, Li Li, Baoguo Li. ✉email: j.speakman@siat.ac.cn

Brown adipose tissue (BAT) primarily evolved as a source of nonshivering thermogenesis that generates endogenous heat, which sustains body temperature in the face of cold exposure[1]. In the past decades, BAT thermogenesis guided by uncoupling protein 1 (UCP1) has been well-characterized and involves allowing protons to bypass complex 5 of the oxidative phosphorylation pathway, and instead move directly from the mitochondrial intermembrane space to the matrix releasing their stored chemiosmotic potential as heat[2]. UCP1 plays a crucial role in preventing fatal hypothermia during acute cold exposure in rodents[3]. However, about 15% of Ucp1-null mice can survive mild cold challenge, suggesting UCP1 is not essential for long-term survival in the cold[4,5]. This led to speculation about the existence of other heat generating mechanisms in Ucp1-null mice, including extended shivering in skeletal muscle[6] or nonshivering UCP1-independent thermogenic mechanisms. Such mechanisms include creatine and SERCA futile pathways[7–11]. The discovery that adult humans possess activated BAT was made using F18-fluorodeoxyglucose positron emission tomography[12–14]. This observation also revealed the importance of BAT in facilitating whole-body glucose homeostasis. Acute cold stimulation[15] or BAT transplantation[16,17] can improve glucose tolerance and insulin sensitivity and these effects are often attributed to activated BAT thermogenesis. However, it is not clear whether the capacity of glucose uptake in BAT can proportionally reflect the nonshivering thermogenesis measured by indirect calorimetry or doubly labeled water (DLW)[18].

The earliest observation suggesting the gut microbiota also participates in fat storage and blood glucose regulation was that germ-free (GF) mice have better glucose homeostasis compared to conventional mice[19]. Later, accumulating studies also demonstrated that depletion of the gut microbiota improves glucose tolerance in obese hosts[20,21]. However, the consequences of microbiota depletion for glucose absorption in different host tissues are currently unknown. It is also unresolved which tissue(s) are mostly responsible for the improved glucose uptake in microbiota-depleted animals. Studies have suggested that browning of white adipose tissue (WAT)[22], hepatic gluconeogenesis[23] or altered cecum enterocyte metabolism[20] may contribute to blood glucose regulation when the gut microbiota is absent. These previous studies reported both antibiotic-treated and germ-free mice have a browning phenotype in their WAT at both room temperature (22 °C) and thermoneutrality (30 °C), and they concluded that the promotion of browning WAT leads to improved glucose tolerance in the microbiota-depleted mice[22]. Although some evidence suggested that browning of WAT increased capacity to combust glucose and fatty acids[24], among the adipose tissues BAT dominates glucose utilization due to the much higher abundance of mitochondria compared with any of the WAT depots[25]. Additionally, our and others' studies have revisited the impact of microbiota depletion on BAT and WAT thermogenesis and found that the process of UCP1-dependent adaptive thermogenesis is impaired in the absence of the gut microbiota[26,27]. A recent study, however, suggested that the gut microbiota effect is an artefact of the way energy expenditure data were analyzed, and that there is no impact on energy expenditure in the cold, once the effect of microbiota depletion on the size of the cecum, and hence elevated contribution of the gut contents to total body mass is taken into account[23]. Nevertheless, this latter study does not distinguish the influence of gut microbiota on UCP1-dependent and UCP1-independent thermogenesis because cold prehabituation can also boost UCP1-independent thermogenesis. They further claimed that gut microbiota-depleted mice do not differ in their performance on the glucose tolerance test or insulin tolerance test, but only have blunted hepatic gluconeogenesis and lower basal blood glucose. However, these conclusions are inconsistent with many earlier studies[19,20,28,29]. These previous studies have focused on WAT or liver as the site for improved glucose homeostasis, but the contribution of BAT in regulating blood glucose after ABX treatment has been largely overlooked. Hence, further investigation is needed to establish whether the improved glucose metabolism in microbiota-deficient mice is real, and if so, if it stems from browning of WAT, hepatic gluconeogenesis or the activation of other tissues such as BAT.

In this work, the main aims are: (1) to re-evaluate the effects of gut microbiota depletion on glucose clearance in mice; (2) to investigate whether the amelioration of glucose clearance was a secondary effect of activated thermogenesis in microbiota-deficient mice; (3) to define the contribution of BAT in modulating glucose uptake in microbiota-deficient mice. We confirm that antibiotic-mediated microbiota depletion improves glucose clearance in mice. We also find that microbiota depletion boosts glucose uptake in BAT and cecum, but neither WAT nor liver, and this response is dissociated from adaptive thermogenesis. Importantly, we show that selective deletion of brown adipocytes mitigates the improvement of glucose clearance in ABX mice.

## Results

**Gut microbiota depletion promotes glucose uptake in brown adipose tissue.** To test the effect of gut microbiota depletion on glucose metabolism in mice, we used the antibiotic cocktail protocol (ABX) used previously, for 3–4 weeks[26], which originated from Abt et al.[30]. Compared to the specific-pathogen-free mice fed with low fat diet (LFD) and housed under conventional conditions (Control), ABX-treated mice exhibited an improvement in glucose clearance in an intraperitoneal glucose tolerance test (ipGTT) at room temperature (22 °C) and cold exposure at 4 °C for 48 h (two-way ANOVA, $F_{3, 36} = 90.55$, $P < 0.001$) (Fig. 1a). Of note, although cold stimulation accelerated glucose clearance in the Control group, it did not further augment glucose clearance in ABX group. Glucose is normally oxidized by mitochondria into water and $CO_2$ through the reactions of the citric acid cycle and oxidative phosphorylation after transport from the circulation into cells. Excess nonmetabolized glucose can also be converted into glycogen or fatty acids when energy demand is attenuated[31]. To determine the state of the glucose mobilization, we conducted a $CO_2$ breath analysis test using $^{13}C$-labeled glucose (Fig. 1b). Similar to our previous [U$^{13}C$] butyrate experiment[26], we intraperitoneally injected a solution containing a mixture of [U$^{13}C$] and $^{12}C$ glucose to fasted Control and ABX mice. Compared to 22 °C, cold stimulation (4 °C) boosted $^{13}CO_2$ production suggesting cold stimulation elevated glucose oxidation. ABX mice had about 1.2-fold higher peak $^{13}CO_2$ output than Control mice at 4 °C (two-way ANOVA, $F_{24, 432} = 9.243$, $P < 0.001$) and no difference was found at 22 °C (Fig. 1c).

To determine the tissues primarily responsible for glucose uptake following depletion of the microbiota, we repeated the same isotope experiment in a second cohort (Fig. 1b). We measured $^{13}C$ content in 22 tissues 25–30 min after injection (at the peak point of $^{13}CO_2$ output). The overall $^{13}C$ glucose uptake was summarized in Fig. 1d. We found cold stimulation enhanced glucose distribution into BAT. At 4 °C, ABX mice had significantly higher $^{13}C$ levels in BAT (two-way ANOVA, $F_{3, 107} = 106.8$, $P < 0.001$), heart (two-way ANOVA, $F_{3, 106} = 6.589$, $P = 0.017$), and liver (two-way ANOVA, $F_{3, 106} = 6.589$, $P = 0.005$) compared to Control mice (Fig. 1e and Supplementary Fig. 1a). At 22 °C, ABX mice had higher $^{13}C$ levels in stomach and colon (two-way ANOVA, $F_{3, 101} = 13.47$, $P = 0.04$) (Supplementary Fig. 1b). There was no difference of glucose uptake in different skeletal muscle depots between Control and ABX mice (two-way ANOVA, $F_{3, 104} = 0.0544$, $P = 0.816$) (Fig. 1f). Upon cold

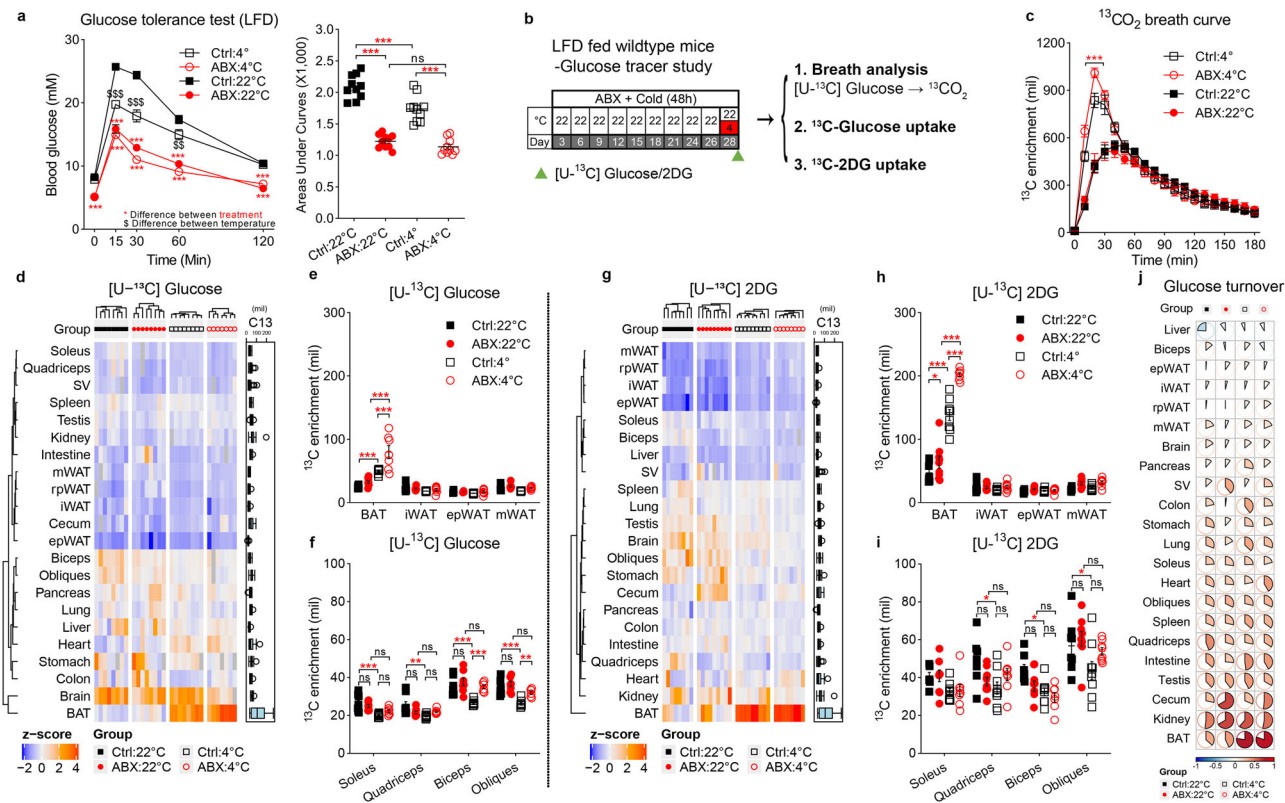

**Fig. 1 Gut microbiota depletion promotes glucose uptake in brown adipose tissue. a** Intraperitoneal glucose tolerance test (ipGTT) and total glucose area under the curves (AUC) ($n = 10$ per group). Control: 22 °C group (black square), ABX:22 °C group (red circle), Control: 4 °C group (black empty square) and ABX:4 °C group (red empty circle) (GTT-Ctrl :22 °C vs ABX:22 °C, $P < 0.001$ at T = 0, 15, 30, 60, 120; GTT-Ctrl :4 °C vs ABX:4 °C, $P = 0.001$ at T = 0, $P < 0.001$ at T = 15, 30, 60, 120; GTT-Ctrl :22 °C vs Ctrl:4 °C, $P < 0.001$ at T = 15, 30, $P = 0.007$ at T = 60. AUC-Ctrl :22 °C vs ABX:22 °C, $P < 0.001$; AUC-Ctrl :4 °C vs ABX:4 °C, $P = 0.001$; AUC-Ctrl :22 °C vs Ctrl:4 °C, $P < 0.001$). **b** Experimental scheme. 4 weeks ABX-treated mice were randomly housed at 22 °C or 48 h acute cold exposure. Then overnight fasted mice were intraperitoneal injected with [U-13C] labelled glucose or 2DG solutions. **c** Glucose metabolized $CO_2$ was determined by 13C enrichment in breath samples at 10 min interval after intraperitoneal injection of [U-13C] labelled glucose at 22 °C or 4 °C ($n = 8$ per group) (T = 30 min, $P < 0.001$). **d** Hierarchical clustering of [U-13C] glucose enrichment (per mil) from 22 tissues based on 13C levels (Ctrl:22 °C, $n = 8$; Ctrl:4 °C, $n = 8$; ABX:22 °C, $n = 8$; ABX:4 °C, $n = 7$). **e, f** 13C enrichment in fat depots (**e**) (BAT: all shown $P < 0.001$) and skeletal muscle (**f**) after injection of [U-13C] labelled glucose (Ctrl:22 °C, $n = 8$; Ctrl:4 °C, $n = 8$; ABX:22 °C, $n = 8$; ABX:4 °C, $n = 7$). **g** Hierarchical clustering of [U-13C] 2DG enrichment (per mil) from 22 tissues (Ctrl:22 °C, $n = 8$; Ctrl:4 °C, $n = 9$; ABX:22 °C, $n = 9$; ABX:4 °C, $n = 8$). **h, i** 13C enrichment in fat depots (**h**) (BAT: Ctrl :22 °C vs ABX:22 °C, $P = 0.013$; BAT: Ctrl :4 °C vs ABX:4 °C, $P < 0.001$; BAT: Ctrl :22 °C vs Ctrl:4 °C, $P < 0.001$; BAT: ABX:22 °C vs ABX:4 °C, $P < 0.001$;) and skeletal muscle (**i**) after injection of [U-13C] labelled 2DG (Ctrl:22 °C, $n = 8$; Ctrl:4 °C, $n = 9$; ABX:22 °C, $n = 9$; ABX:4 °C, $n = 8$). **j** 13C glucose turnover rate in four groups using an exponential uptake-elimination model. All statistical analyses were performed by two-way ANOVA with Bonferroni's multiple comparisons. All results are given as mean ± SEM. Differences with $P < 0.05$ were considered to be significant. $P < 0.05$ (* or $), $P < 0.01$ (** or $$), and $P < 0.001$ (*** or $$$). SV seminal vesicle, iWAT inguinal WAT, epWAT epididymal WAT, mWAT mesenteric WAT, rpWAT retroperitoneal WAT. See also Supplementary Fig. 1.

stimulation, Control mice decreased glucose uptake into skeletal muscles whereas ABX mice did not (Fig. 1f). There was no difference in 13C enrichment between Control and ABX mice in inguinal, epididymal, mesenteric or retroperitoneal WAT at both 22 °C and 4 °C (two-way ANOVA, $F_{3,80} = 1.426$, $P = 0.239$) (Fig. 1e and Supplementary Fig. 1c). The 13C levels in the remaining tissues was unchanged after ABX treatment (Supplementary Fig. 1a–c). Consistent with our previous study[26], ABX mice had lower white fat mass from different anatomic sites (Supplementary Fig. 1d–g).

The 13C glucose uptake experiment reflected the net levels of glucose content in different tissues 30 min after glucose injection. That is the difference between uptake and utilization. To further confirm the redistribution of glucose in the microbiota-depleted mice focusing only on uptake, we injected a mixture of [U13C] 2-Deoxy-D-glucose (2DG) and 12C glucose solution at the same ratio as in the 13C glucose experiment to fasted Control and ABX mice (Fig. 1b).

Although the glucose transporters can import 2DG into cells, the modification of 2-hydroxyl group in 2DG prevents its degradation[32]. Hence, using isotope labeled 2DG allowed us to measure the cumulative glucose uptake from the time of injection to the peak point, independent of utilization. The overall 13C 2DG uptake was summarized in Fig. 1g. At 4 °C, similar to the 13C glucose experiment, ABX mice had significant higher 13C-labeled 2DG in BAT (two-way ANOVA, $F_{3,29} = 29.00$, $P < 0.001$) and heart (two-way ANOVA, $F_{3,30} = 5.304$, $P = 0.017$) (Fig. 1h and Supplementary Fig. 1h). Cold stimulation decreased 13C levels in skeletal muscles in Control mice (two-way ANOVA, $F_{1,59} = 24.92$, $P < 0.001$) but not in ABX mice (two-way ANOVA, $F_{1,59} = 3.832$, $P = 0.055$) (Fig. 1i). At 22 °C, ABX mice had higher 13C-labeled 2DG in BAT (two-way ANOVA, $F_{3,29} = 29.00$, $P = 0.013$) and cecum (two-way ANOVA, $F_{3,30} = 63.84$, $P < 0.001$) (Fig. 1h and Supplementary Fig. 1i). We did not observe any difference in cumulative 13C levels in the rest of the tissues including all WAT depots and liver after ABX treatment

(Supplementary Fig. 1h and Supplementary Fig. 1j). Different to the [13]C glucose experiment, the [13]C 2DG level in the liver did not show a difference. This could be because 2DG only reflects glucose uptake while the glucose tracer reflected glucose uptake and utilization. Notably, the capacity of basal glucose utilization in all WAT depots was trivial compared to other tissues and ABX treatment failed to enhance the glucose turnover, whereas ABX mice had higher glucose turnover in BAT and cecum (Fig. 1j).

Overall, we found that ABX treatment accelerated glucose clearance and our two isotope tracer experiments showed that ABX treatment only increased glucose uptake and utilization in BAT and tissues of the alimentary tract. In ABX mice, cold exposure further stimulated glucose uptake in BAT and heart. We did not detect differences in [13]C levels in all WAT depots (inguinal, epididymal, mesenteric, or retroperitoneal WAT) and liver. These data indicated that BAT, heart, and the alimentary tract, rather than WAT depots or liver, are the most important sites for enhanced glucose disposal when gut microbiota was absent.

**Gut microbiota is not required for UCP1-independent thermogenesis.** ABX treatment leads to a reduction of UCP1 expression and impairment of UCP1-dependent adaptive thermogenesis in low fat diet (LFD) fed wild-type mice[26]. Since UCP1 is dispensable for long-term thermoregulation but essential for survival in the acute cold challenge, it is important to distinguish the effect of gut microbiota depletion on UCP1-dependent and UCP1-independent thermogenesis. To this end, we treated LFD-fed Ucp1-KO mice and their wild-type siblings with the ABX cocktail for 4–5 weeks at room temperature (Fig. 2a and Supplementary Fig. 2a). As in previous studies[3,33,34], we found that Ucp1-KO mice had similar body weight compared to sibling controls at 22 °C. Additionally, we also found 4 weeks of ABX treatment did not significantly lower their body weight (Supplementary Fig. 2b). Magnetic resonance spectroscopy (MRS) analyses revealed that ABX treatment lowered total body fat in wild-type mice (two-way ANOVA, $F_{3,\ 64} = 2.571$, $P < 0.001$) but not in Ucp1-KO mice (two-way ANOVA, $F_{3,\ 64} = 2.571$, $P = 0.165$) (Fig. 2b). We assessed the glucose clearance in ipGTT at the 3rd week of ABX treatment. Although control treated Ucp1-KO mice had similar body weight and total body fat to the control treated wild-type mice, we found that control treated Ucp1-KO mice had impaired glucose clearance compared with wild-type mice (one-way ANOVA, $F_{3,\ 33} = 4.261$, $P = 0.003$). After ABX treatment, both Ucp1-KO and wild-type mice had significantly improved glucose clearance (one-way ANOVA, $F_{3,\ 33} = 4.261$, $P < 0.001$) (Fig. 2c, d).

Although we previously reported that the gut microbiota is required for UCP1-dependent thermogenesis of BAT in wild-type mice[26], a recent study has challenged this notion and reported the gut microbiota is dispensable to β3-adrenoceptor activation in the commensal-depleted (CD) mice[23]. Despite the difference in the experimental animals, 1 h stimulation of CL-316243 is insufficient to evaluate the activation of thermogenesis. To address these conflicting results, we repeated our experiment at 4.5 weeks of ABX treatment with a shorter data collection interval (6 min per data point) than our previous measurement, which allows us to observe more detailed information using indirect calorimetry (TSE PhenoMaster). As the response to CL-316243 stimulation, both Control and ABX-treated wild-type mice immediately elevated their oxygen consumption. The oxygen consumption reached a plateau after about 6 h and further increased as the mice entered the dark phase. The oxygen consumption did not differ significantly between Control and ABX-treated wild-type mice over the first hours (ANCOVA, $F_{1,\ 11} = 0.02$, $P = 0.8901$),

which was consistent with the result in Krisko et al., but ABX-treated wild-type mice failed to maintain the high oxygen consumption rate and significantly dropped by about 10% after 6 h of CL-316243 stimulation (ANCOVA, $F_{1,\ 11} = 3.36$, $P = 0.033$) (Fig. 2e, f). Since UCP1 is essential for adaptive adrenergic nonshivering thermogenesis[35,36], both Control and ABX-treated Ucp1-KO mice did not respond to CL-316243 stimulation. The activation of β3-adrenoceptor leads to lipolysis, which was evident in respiratory exchange ratio (RER). Following CL-316243 injection, RER declined from 0.9 to less than 0.8 in the four groups of mice indicating that fat was being increasingly used for energy combustion. A persistent low RER in Control treated wild-type mice was observed where the RER in other three groups regained their basal levels after dark phase (two-way ANOVA, $F_{3,\ 27} = 21.00$, $P < 0.001$) (Fig. 2g, h). We did not observe any difference in physical activity among four groups during the light phase (two-way ANOVA, $F_{3,\ 27} = 3.396$, $P > 0.200$) (Supplementary Fig. 2c, d).

We next asked whether gut microbiota depletion affected UCP1-independent thermogenic mechanisms. We assessed energy expenditure in ABX-treated Ucp1-KO mice after a 10 days step-down cooling protocol (Fig. 2i, also see Methods for details). Microbiota depletion did not affect body weight (two-way ANOVA, $F_{3,\ 31} = 0.6147$, $P = 0.611$) in Ucp1-KO mice after 3–4 weeks of ABX treatment (Supplementary Fig. 2e). Cold adapted Ucp1-KO mice had elevated food intake compared to mice at 22 °C (2-way ANOVA, $F_{3,\ 31} = 10.46$, $P < 0.001$) (Supplementary Fig. 2f). At 22 °C, ABX-treated Ucp1-KO mice displayed comparable levels of total body fat and lean mass (two-way ANOVA, $F_{3,\ 62} = 2.689$, $P > 0.999$) (Fig. 2j and Supplementary Fig. 2g–k). Although the acute cold exposure stimulates lipolysis, 10 days cold habituation increased the total fat mass compared to that at room temperature. At 22 °C, ABX-treated Ucp1-KO mice did not have significantly altered daily energy expenditure (two-way ANOVA, $F_{1,\ 13} = 0.6596$, $P = 0.431$) (Fig. 2k). To measure the energy metabolism of cold habituated mice at 4 °C, we used the doubly labeled water (DLW) technique[37]. Compared to 22 °C, cold exposure elevated energy expenditure in both Control and ABX-treated Ucp1-KO mice, which reflected the activation of UCP1-independent thermogenesis. Importantly, ABX-treated Ucp1-KO mice did not differ in their daily energy expenditure from Control Ucp1-KO mice at 4 °C (one-way ANOVA, $F_{3,\ 29} = 0.5414$, $P > 0.999$) indicating microbiota depletion had minimal impact on UCP1-independent adaptive thermogenesis (Fig. 2l). We repeated these analyses using the approach suggested by Krisko et al.[23] employing our previous dissection data to estimate the weight of the cecum contents. We then subtracted these weights from the total live mass before running the analysis. In the animals at room temperature, we found no effect of the adjusted body weight (ANCOVA, $F_{1,12} = 1.09$, $P = 0.318$) and no effect of ABX treatment (ANCOVA, $F_{1,12} = 0.43$, $P = 0.526$). In the cold exposed groups, we found a significant effect of the adjusted weight (ANCOVA: $F_{1,12} = 6.83$, $P = 0.02$) but no ABX treatment effect (ANCOVA, $F_{1,12} = 1.54$, $P = 0.234$). In other words, using the analysis approach advocated by Krisko et al. (2020) made no difference to the outcome that ABX treatment had no impact on energy expenditure when UCP1 was absent.

**UCP1 is dispensable for gut microbiota depletion-induced glucose improvement.** Numerous studies have suggested that elevated UCP1 levels in BAT have a positive correlation with raised metabolic rate and better glucose metabolism in both rodents and humans[38]. Previous studies have linked the beneficial effect of glucose metabolism in ABX mice to the elevated *Ucp1*

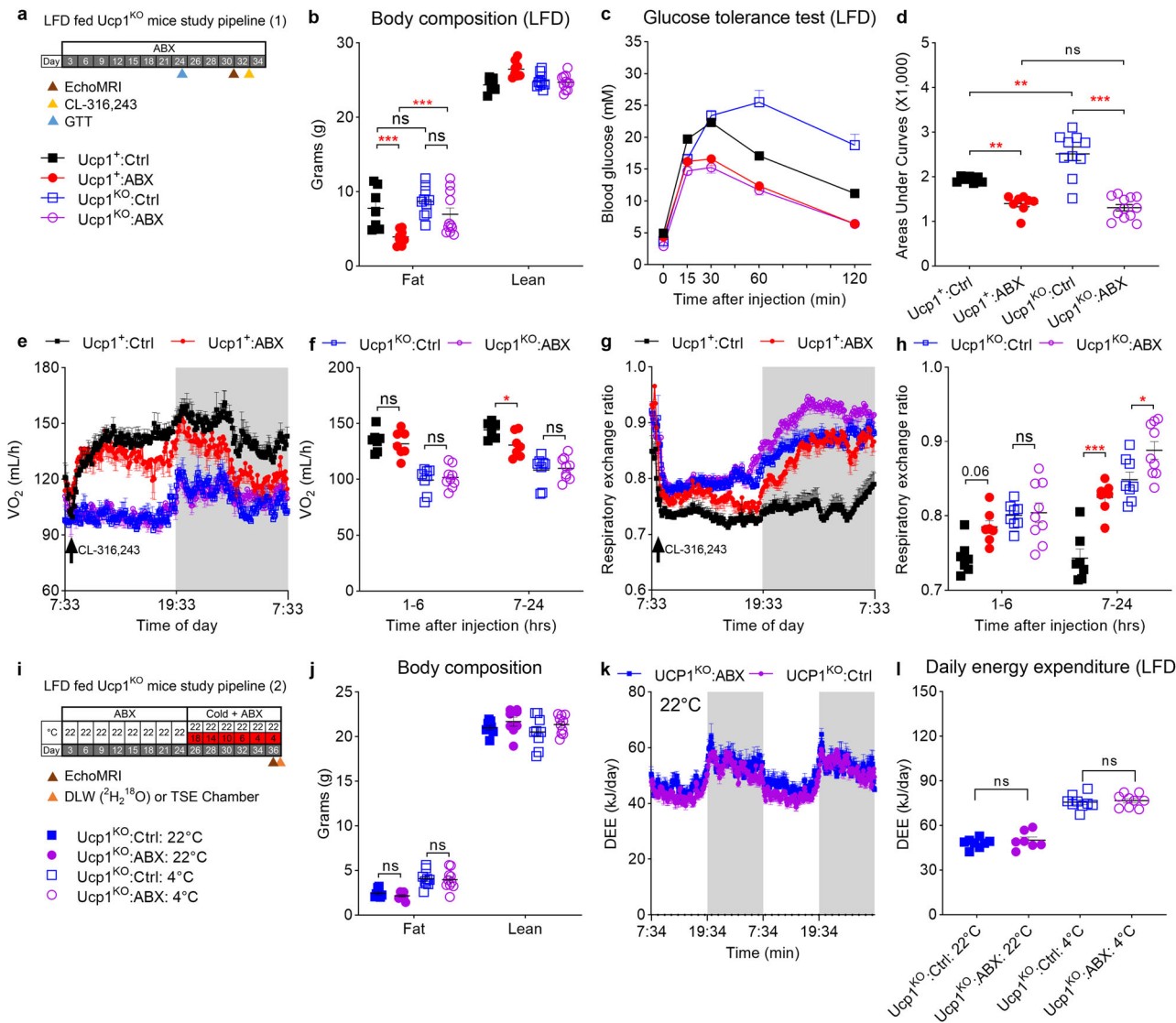

**Fig. 2 Gut microbiota is not required for UCP1-independent thermogenesis. a** Experimental scheme. 16-weeks low fat diet (LFD)-fed Ucp1-KO mice and its sibling controls subject to ABX treatment. Ucp1+: Ctrl group (black square) and Ucp1KO: Ctrl group (blue empty square) were control groups, Ucp1+: ABX group (red circle) and Ucp1KO: ABX group (purple empty circle) received 4–5 weeks of ABX treatment (Ucp1+: Ctrl, $n = 7$; Ucp1+: ABX, $n = 8$; Ucp1KO: Ctrl, $n = 10$; Ucp1KO: ABX, $n = 12$). The results shown are representative of two independent experiments. **b** Body composition at fourth week of ABX treatment (Ucp1+: Ctrl, $n = 7$; Ucp1+: ABX, $n = 8$; Ucp1KO: Ctrl, $n = 10$; Ucp1KO: ABX, $n = 11$) (both $P < 0.001$). **c, d** Intraperitoneal glucose tolerance test (ipGTT) and total glucose area under the curves (AUC) at the third week of ABX treatment (Ucp1+: Ctrl, $n = 7$; Ucp1+: ABX, $n = 8$; Ucp1KO: Ctrl, $n = 10$; Ucp1KO: ABX, $n = 12$) (**d**: Ucp1+: Ctrl vs Ucp1+: ABX, $P = 0.009$; Ucp1+: Ctrl vs Ucp1KO: Ctrl, $P = 0.003$; Ucp1KO: Ctrl vs Ucp1KO: ABX, $P < 0.001$). **e–h** The curve of oxygen consumption (**e**) and respiratory exchange ratio (**g**) after CL-316246 injection, and the average of oxygen consumption (**f**) and respiratory exchange ratio ($P = 0.033$) (**h**) (Ucp1+: Ctrl, $n = 7$; Ucp1+: ABX, $n = 8$; Ucp1KO: Ctrl, $n = 8$; Ucp1KO: ABX, $n = 9$) (Ucp1+, $P < 0.001$; Ucp1KO, $P = 0.002$). **I** Experimental scheme. LFD-fed Ucp1-KO mice subject to gradient cold stimulation and ABX treatment. Ucp1KO:Ctrl:22 °C (blue square) group was control; Ucp1KO:Ctrl:4 °C (blue empty square) group habituated to 10 days of gradient cold stimulation; Ucp1KO:ABX:22 °C group (purple circle) group received ABX treatment for 36 days; and Ucp1KO:ABX:4 °C (purple empty circle) group habituated to 10 days of gradient cold stimulation at 24th day of ABX treatment until day 36 (Ucp1KO:Ctrl:22 °C, $n = 8$; Ucp1KO:Ctrl:4 °C, $n = 9$; Ucp1KO:ABX:22 °C, $n = 8$; Ucp1KO:ABX:4 °C, $n = 10$). **J** Final body composition (Ucp1KO:Ctrl:22 °C, $n = 8$; Ucp1KO:Ctrl:4 °C, $n = 9$; Ucp1KO:ABX:22 °C, $n = 8$; Ucp1KO:ABX:4 °C, $n = 10$). **k, l** Daily energy expenditure at room temperature (**k**) (Ucp1KO:Ctrl:22 °C, $n = 8$; Ucp1KO:Ctrl:4 °C, $n = 9$; Ucp1KO:ABX:22 °C, $n = 7$; Ucp1KO:ABX:4 °C, $n = 9$) or cold habituation (**l**) (Ucp1KO:Ctrl, $n = 8$; Ucp1KO:ABX, $n = 7$;) in LFD-fed Ucp1-KO mice. All statistical analyses were performed by two-way ANOVA with Bonferroni's multiple comparisons. All results are given as mean ± SEM. Differences with $P < 0.05$ were considered to be significant. $P < 0.05$ (*), $P < 0.01$ (**), and $P < 0.001$ (***). See also Supplementary Fig. 2.

mRNA expression and browning of inguinal and epididymal WAT[22]. Contradicting these observations, we found that both mRNA and protein levels of UCP1 in BAT and inguinal WAT were decreased, whereas the expression of *Ucp1* mRNA was unchanged in cold stimulated ABX mice[26]. In this study, we also observed glucose uptake to all depots of white adipose tissue was

unchanged in ABX mice, at room temperature and in cold condition (Fig. 1d). To further evaluate whether UCP1 is essential for the beneficial effect of gut microbiota depletion on glucose clearance in obese mice, we administered ABX to 60% kcal high-fat diet (HFD) (D12492, Research Diets, New Brunswick, NJ) fed Ucp1-KO mice and their wild-type siblings for 4–5 weeks

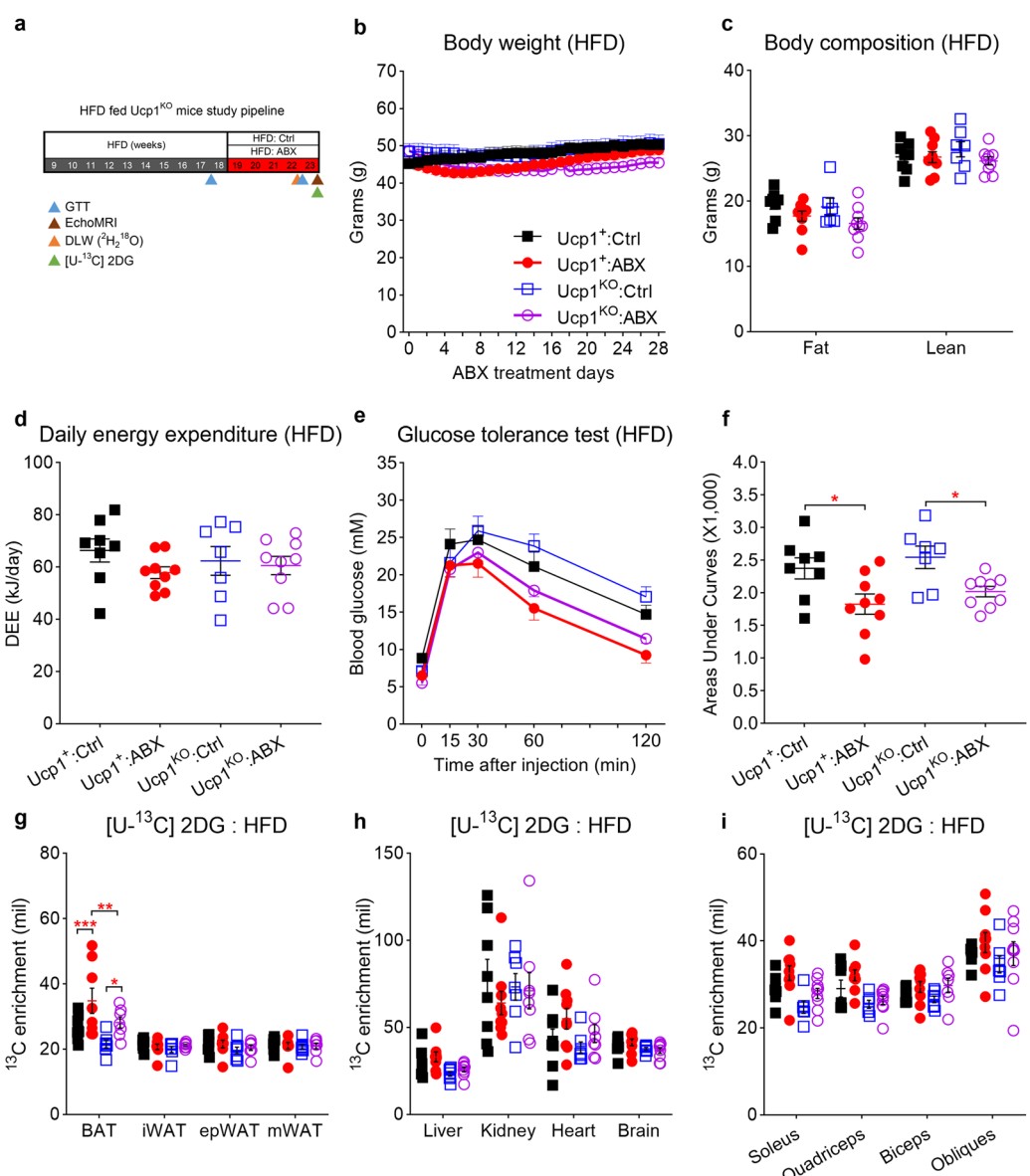

**Fig. 3 UCP1 is dispensable for gut microbiota depletion-induced glucose improvement. a** Experimental scheme. 10-weeks high-fat diet (HFD)-fed Ucp1-KO mice and its sibling controls subject to ABX treatment. Ucp1+:Ctrl group (black square) and Ucp1KO:Ctrl group (blue empty square) were Control groups, Ucp1+:ABX group (red circle) and Ucp1KO:ABX group (purple empty circle) received 5 weeks of ABX treatment (Ucp1+: Ctrl, $n = 9$; Ucp1+: ABX, $n = 9$; Ucp1KO: Ctrl, $n = 7$; Ucp1KO: ABX, $n = 9$). The results shown are representative of two independent experiments. **b–d** Body mass curves (**b**), final body composition (**c**), and daily energy expenditure (**d**) of HFD-fed Ucp1-KO mice and its sibling at Control or ABX condition (Ucp1+: Ctrl, $n = 8$; Ucp1+: ABX, $n = 9$; Ucp1KO: Ctrl, $n = 7$; Ucp1KO: ABX, $n = 9$). **e, f** Intraperitoneal glucose tolerance test (ipGTT) and total glucose area under the curves (AUC) at the fourth week of ABX treatment (Ucp1+: Ctrl, $n = 9$; Ucp1+: ABX, $n = 9$; Ucp1KO: Ctrl, $n = 7$; Ucp1KO: ABX, $n = 9$) (Ucp1+, $P = 0.011$; Ucp1KO, $P = 0.017$). **g–i** 2DG enrichment in fat depots (**g**), skeletal muscle (**i**), and other tissues (**h**) after intraperitoneal injection of [U-13C] labelled 2DG to mice at the 5th week of ABX (Ucp1+: Ctrl, $n = 8$; Ucp1+: ABX, $n = 9$; Ucp1KO: Ctrl, $n = 7$; Ucp1KO: ABX, $n = 7$) (BAT, Ucp1+: Ctrl vs Ucp1+: ABX, $P < 0.001$; Ucp1+: ABX vs Ucp1KO: ABX, $P = 0.008$; Ucp1KO: Ctrl vs Ucp1KO: ABX, $P = 0.012$). All statistical analyses were performed by two-way ANOVA with Bonferroni's multiple comparisons. All results are given as mean ± SEM. Differences with $P < 0.05$ were considered to be significant. $P < 0.05$ (*), $P < 0.01$ (**), and $P < 0.001$ (***). See also Supplementary Fig. 3.

(Fig. 3a). Five weeks of ABX treatment at 22 °C did not affect body weight (two-way ANOVA, $F_{3, 29} = 1.62$, $P = 0.206$) or food intake (two-way ANOVA, $F_{3, 29} = 2.161$, $P = 0.114$) in HFD-fed wild-type and Ucp1-KO mice (Fig. 3b and Supplementary Fig. 3a). ABX treatment also did not affect total body fat mass in HFD-fed Ucp1-KO mice (two-way ANOVA, $F_{3, 57} = 2.434$, $P = 0.336$) or wild-type mice (two-way ANOVA, $F_{3, 57} = 2.434$, $P = 0.859$) (Fig. 3c). There was no difference in tissue weight except the enlarged intestine (two-way ANOVA, $F_{3, 68} = 11.48$, $P < 0.001$) and cecum (two-way ANOVA, $F_{3, 68} = 11.48$, $P < 0.001$)

after ABX treatment (Supplementary Fig. 3b–e). In the control condition at 22 °C, consistent with previous studies[33], HFD-fed wild-type and Ucp1-KO mice had comparable daily energy expenditure (ANCOVA, $F_{3, 28} = 1.126$, $P = 0.37$). Microbiota depletion also did not alter daily energy expenditure in wild-type and Ucp1-KO mice (one-way ANOVA, $F_{3, 29} = 1.126$, $P = 0.475$) (Fig. 3d). To better understand the effect of microbiota depletion on glucose metabolism, baseline ipGTT was performed 1 week before ABX treatment and a second ipGTT was conducted on the fourth week of ABX treatment. In the baseline state, although

there was no difference in body weight or body fat, HFD-fed Ucp1-KO mice had slower glucose clearance performance compare to wild-type mice (two-way ANOVA, $F_{1, 29} = 6.15$, $P = 0.019$) (Supplementary Fig. 3f, g). After ABX treatment, both HFD-fed wild-type (two-way ANOVA, $F_{12, 120} = 2.951$, $P < 0.036$) and Ucp1-KO mice (two-way ANOVA, $F_{12, 120} = 2.951$, $P < 0.017$) had significantly accelerated glucose clearance at 60 min and 120 min (Fig. 3e) and their total glucose area under the curve (AUC) was significantly lowered after ABX treatment (one-way ANOVA, $F_{3, 30} = 4.831$, $P = 0.016$) (Fig. 3f).

To determine the impact of ABX on glucose uptake in obese Ucp1-KO mice, a $^{13}$C-labeled 2DG solution was used. Similar to LFD-fed mice in Supplementary Fig. 1i, ABX treatment increased $^{13}$C levels in the cecum of both Ucp1-KO mice and wild-type controls (two-way ANOVA, $F_{3,29} = 56.54$, $P = 0.01$) (Supplementary Fig. 3h). ABX treatment also significantly elevated $^{13}$C 2DG uptake to BAT in obese wild-type (two-way ANOVA, $F_{3, 26} = 4.00$, $P < 0.001$) and Ucp1-KO (two-way ANOVA, $F_{3, 26} = 4.00$, $P = 0.012$) groups (Fig. 3g). Notably, the obese Ucp1-KO group had significant lower $^{13}$C 2DG contents in BAT compared to wild-type group when microbiota was depleted. 2DG contents in all white adipose depots and remaining tissues including liver had no changes (Fig. 3g–i and Supplementary Fig. 3i). Together, our data indicated that at room temperature microbiota depletion improved glucose clearance in both lean and obese mice, and ABX treatment increased the glucose uptake in BAT and cecum. This action was independent of total body weight, daily food intake or the expression of UCP1.

**Brown and beige adipocytes are indispensable for gut microbiota depletion-induced glucose improvement.** In the glucose tracer experiments, we found glucose uptake to both the BAT (Figs. 1e and 3g) and cecum (Supplementary Figs. 1f and 3h) was significantly increased. It has been documented that the capacity to utilize glucose in BAT is positively associated with improvement of glucose tolerance in both rodents and humans[15,39]. A recent study found the cecum shifted energy utilization from short-chain fatty acids to glucose when the microbiota was absent[20]. To probe whether cells expressing UCP1 (Ucp1$^+$) are necessary for the glucose amelioration in ABX treatment, we selectively deleted Ucp1$^+$ cells in LFD-fed Ucp1$^{DTR}$ mice by injection of diphtheria toxin (DT) during ABX treatment at 22 °C. Ucp1$^{DTR}$ mice have a diphtheria toxin receptor (DTR) and an EGFP cassette under the control of the *Ucp1* promoter[40], which provides an inducible way to delete Ucp1$^+$ cells without any developmental deficit[41,42]. LFD-fed Ucp1$^{DTR}$ mice were randomly divided into four groups: (1) Saline-Ctrl group, which received subcutaneous saline injections and drank autoclaved distilled water; (2) DT-Ctrl group, which were given subcutaneous DT injections and drank autoclaved distilled water; (3) Saline-ABX group, which received saline injections and were administrated ABX in the autoclaved distilled water; and (4) DT-ABX group, which were given subcutaneous DT injections and ABX treatment (Fig. 4a and Supplementary Fig. 4a). Neither depletion of microbiota nor deletion of Ucp1$^+$ cells had an impact on body weight in the Ucp1$^{DTR}$ mice (two-way ANOVA, $F_{3, 19} = 1.860$, $P = 0.171$) (Fig. 4b). DT injection significantly lowered the BAT weight in DT-Ctrl and DT-ABX groups (one-way ANOVA, $F_{3, 19} = 1.546$, $P < 0.001$) (Supplementary Fig. 4d–g). After 3 weeks of ABX and DT experiment, the Saline-ABX treatment group showed an improvement compared to the Saline-Ctrl group in ipGTT (two-way ANOVA, $F_{3, 19} = 8.91$, $P < 0.001$). Noteworthy, the combination of DT and ABX treatment neutralized the beneficial effect of ABX in ipGTT (Fig. 4c, d),

showing Ucp1$^+$ cells are required for the ABX-mediated improvement in the ipGTT. Both ABX treatment groups (Saline-ABX and DT-ABX) had a reduction of the oxygen consumption during the night (two-way ANOVA, $F_{3, 19} = 4.208$, $P = 0.015$). We did not observe any difference in RER (Fig. 4g, h) and physical activity among the four groups (Supplementary Fig. 4b, c).

To further investigate whether Ucp1$^+$ cells are essential for the ABX-mediated glucose clearance in obese mice. We fed Ucp1$^{DTR}$ mice with 60% kcal high-fat diet for 9–10 weeks (Fig. 5a), following five weeks of ABX treatment at 22 °C. In the obese Ucp1$^{DTR}$ mice, neither depletion of microbiota nor deletion of Ucp1$^+$ cells had an impact on body weight (two-way ANOVA, $F_{3, 22} = 0.7478$, $P = 0.535$) (Fig. 5b), daily food intake (two-way ANOVA, $F_{3, 22} = 1.061$, $P = 0.386$) (Supplementary Fig. 5a), or body composition (two-way ANOVA, $F_{3, 50} = 4.598$, $P = 0.386$) (Fig. 5c) after 4 weeks treatment. Consistent with ABX-treated obese Ucp1-KO mice, the intestine (two-way ANOVA, $F_{3, 96} = 212$, $P < 0.001$) and cecum (two-way ANOVA, $F_{3, 96} = 212$, $P = 0.047$) were heavier after ABX treatment and the weights of other tissues remained unchanged (Supplementary Fig. 5b–e). A recent study suggested Ucp1$^+$ cells do not contribute to energy expenditure at HFD-fed state[42]. Confirming this finding, we also found that Ucp1$^+$ cells had no impact on energy expenditure when the microbiota was depleted (ANCOVA, $F_{3, 15} = 0.5$, $P = 0.69$) (Fig. 5d). Before the ABX treatment was started, baseline ipGTT (two-way ANOVA, $F_{3, 28} = 0.11$, $P = 0.953$) and AUC (one-way ANOVA, $F_{3, 28} = 0.26$, $P = 0.26$) were not significantly different between the four groups (Supplementary Fig. 5f, g). After 4 weeks of ABX and DT experiment, compared to the Saline-Ctrl group, the DT-Ctrl group (two-way ANOVA, $F_{3, 28} = 6.12$, $P = 0.048$) had deteriorated ipGTT performance, whereas the Saline-ABX treatment group showed an improvement (two-way ANOVA, $F_{3, 28} = 6.12$, $P = 0.037$). Noteworthy, the combination of DT and ABX treatment neutralized the beneficial effect of ABX on ipGTT (Fig. 5e, f), showing Ucp1$^+$ cells are necessary for the ABX-mediated improvement on ipGTT.

To determine the influence of depletion of Ucp1$^+$ cells on the uptake of glucose across tissues when the microbiota was deficient, we injected $^{13}$C-labeled 2DG solution at the 5th week of the experiment. Consistent with the $^{13}$C 2DG result in ABX-treated obese wild-type mice at Supplementary Fig. 3h, the Saline-ABX group (two-way ANOVA, $F_{3, 24} = 6.82$, $P < 0.001$) and DT-ABX group (two-way ANOVA, $F_{3, 24} = 6.82$, $P < 0.001$) both had increased $^{13}$C 2DG contents in cecum compared to the two non-ABX-treated groups (Supplementary Fig. 5h). Similar to the result in Fig. 3g, the Saline-ABX group had enhanced 2DG levels in BAT compared to the Saline-Ctrl group (two-way ANOVA, $F_{3, 89} = 18.12$, $P = 0.027$) and unchanged 2DG levels in both WAT and liver (Fig. 5g). Both the DT-Ctrl group and DT-ABX group (two-way ANOVA, $F_{3, 89} = 18.12$, $P < 0.001$) had significantly impaired $^{13}$C 2DG uptake in BAT, demonstrating that depletion of Ucp1$^+$ cells impaired glucose uptake to BAT (Fig. 5g). 2DG contents in heart also decreased following DT injection (two-way ANOVA, $F_{3, 90} = 1.513$, $P < 0.001$). DT injection did not affect $^{13}$C 2DG contents in all the white adipose depots (two-way ANOVA, $F_{3, 69} = 0.4416$, $P = 0.724$) suggesting that the depletion of Ucp1$^+$ beige adipocytes had no effect to 2DG uptake in ABX-treated mice (Fig. 5g). The remaining tissues including liver (two-way ANOVA, $F_{3, 24} = 0.1213$, $P = 0.947$) and muscle had no difference in 2DG uptakes (Fig. 5h, i and Supplementary Fig. 5i). These data indicated that BAT, rather than liver or WAT depots (e.g., inguinal or epididymal WAT), is the key organ in modulating glucose clearance when the gut microbiota was depleted.

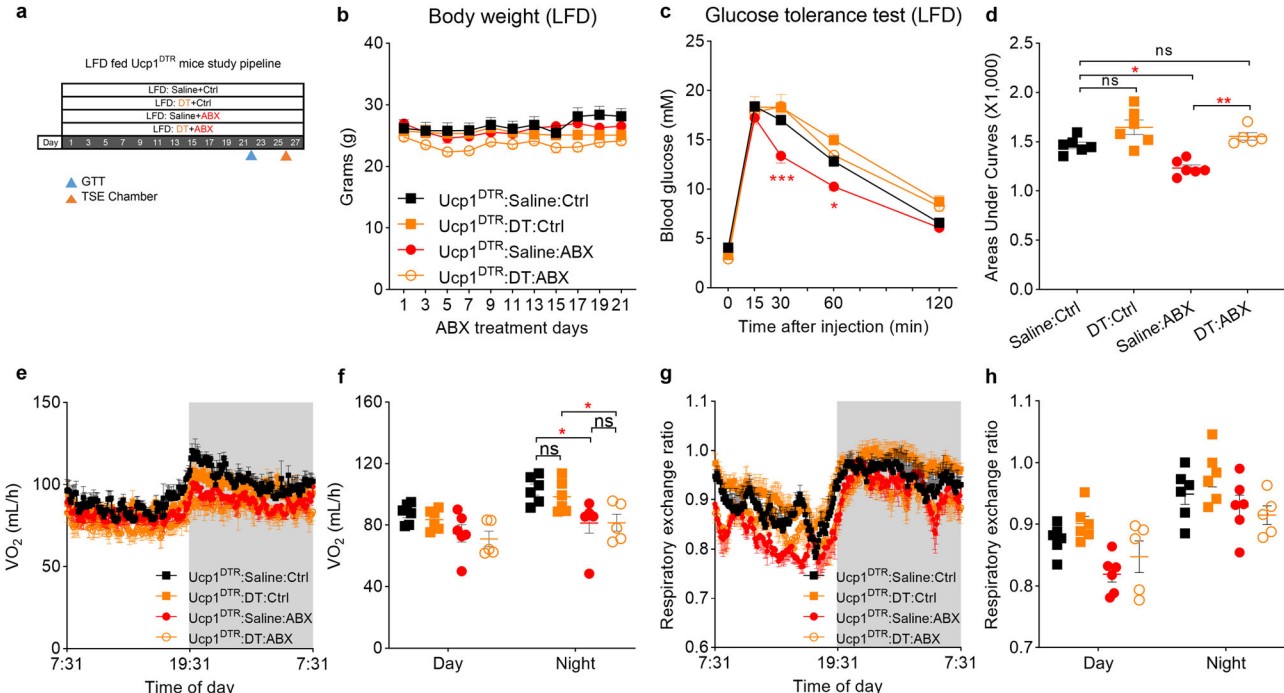

**Fig. 4 Deletion of Ucp1+ cells in lean mice blocked the gut microbiota depletion-modulated glucose clearance improvement. a** Experimental scheme. 10-weeks low fat diet (LFD)-fed Ucp1DTR mice were divided into four groups. Saline-Ctrl group (black square) received subcutaneous saline injections and drank autoclaved water; DT-Ctrl group (orange square) given subcutaneous DT injection and drank autoclaved water; Saline-ABX group (red circle) received saline injections and ABX treatment; and DT-ABX group (orange open circle) given subcutaneous DT injection and ABX treatment (Ucp1DTR: Saline:Ctrl, n = 6, Ucp1DTR:DT:Ctrl, n = 6; Ucp1DTR:Saline:ABX, n = 6; Ucp1DTR:DT:ABX, n = 6). The results shown are representative of two independent experiments. **b** Body mass curves of four different treated HFD-fed Ucp1-DTR mice (Ucp1DTR:Saline:Ctrl, n = 6, Ucp1DTR:DT:Ctrl, n = 6; Ucp1DTR:Saline: ABX, n = 6; Ucp1DTR:DT:ABX, n = 5). The results shown are representative of two independent experiments. **c, d** Intraperitoneal glucose tolerance test (ipGTT) and total glucose area under the curves (AUC) at the third week of treatment (Ucp1DTR:Saline:Ctrl, n = 6, Ucp1DTR:DT:Ctrl, n = 6; Ucp1DTR:Saline: ABX, n = 6; Ucp1DTR:DT:ABX, n = 5) (**c**: Ucp1DTR:Saline:Ctrl vs Ucp1DTR:Saline:ABX, P < 0.001 at T = 30 min, P = 0.016 at T = 60 min; **d**: Saline:Ctrl vs Saline:ABX, P = 0.017, Saline:ABX vs DT:ABX, P = 0.001). **e–h** The curve of oxygen consumption (**e**) and respiratory exchange ratio (**g**) in 24 h, and the average of oxygen consumption (**f**) and respiratory exchange ratio (Ucp1DTR:Saline:Ctrl vs Ucp1DTR:Saline:ABX, P = 0.011; Ucp1DTR:DT:Ctrl vs Ucp1DTR:DT: ABX, P < 0.018) (**h**) in day and night states (Ucp1DTR:Saline:Ctrl, n = 6, Ucp1DTR:DT:Ctrl, n = 6; Ucp1DTR:Saline:ABX, n = 6; Ucp1DTR:DT:ABX, n = 5). DT diphtheria toxin, DTR diphtheria toxin receptor. All statistical analyses were performed by two-way ANOVA with Bonferroni's multiple comparisons. All results are given as mean ± SEM. Differences with P < 0.05 were considered to be significant. P < 0.05 (*), P < 0.01 (**), and P < 0.001 (***). See also Supplementary Fig. 4.

## Discussion

Increasing numbers of studies have established a connection between the gut microbiota and host metabolism in both mice and humans[29,43–45]. Although one recent study suggested microbiome-deficient mice exhibited no differences in tolerance to glucose[23], many previous studies have found that microbiota depletion, using either antibiotic treatment or GF models, improves glucose tolerance[19,20,22,46]. Our first experiment was to reconfirm whether microbiota deficiency causes improvement in the ipGTT in wild-type mice. We found that microbiota depletion accelerated glucose clearance from circulation, and ABX mice had higher $^{13}C$ $CO_2$ peak output at 4 °C after $^{13}C$ glucose injection indicating the mice preferred to metabolize the injected energy substrate when the microbiota was absent. Our isotope tracing experiments evaluated $^{13}C$-labeled glucose and 2DG contents in 22 tissues in Control mice and this indicated that cold exposure only stimulated uptake of glucose in BAT. This reflects the well-established physiological acclimation response, whereby rodents shift the source of heat production from shivering in skeletal muscles to nonshivering thermogenesis (NST) in BAT after 48 h cold stimulation[47]. It is noteworthy that the glucose uptake in skeletal muscle was not decreased in ABX mice after acute cold exposure indicating a possible extension of shivering in ABX

mice[48]. A previous study linked improvement in glucose metabolism in ABX mice to elevation of glucose uptake in inguinal WAT[22], although uptake to other tissues was not studied. Here, we found that BAT had greatly enhanced glucose uptake while all the WAT depots had negligible changes when the microbiota was absent (Fig. 1e, h). Since it was already characterized[22,23,26] that the microbiota-deficient mice have lower body fat mass compare to Control mice under LFD conditions (Supplementary Fig. 1d), the total contribution of WAT depots to glucose uptake is even lower in ABX-treated mice.

Importantly, ABX mice not only had higher glucose uptake in BAT, but also showed elevated glucose uptake in the digestive organs, suggesting these tissues are important for glucose uptake. Depletion of Ucp1+ cells fully blocked the beneficial effect of ABX in ipGTT indicating that BAT is required for the improved glucose clearance when the gut microbiota was absent. We did not determine whether the cecum is also necessary for this action in this study. Although a recent study addressed ABX treatment mice with the cecum removed, cecum removal only partially blunted the effect of ABX on glucose clearance[23]. The liver is an important organ for the glycogen synthesis. In current study, we mainly focused on glucose uptake and glucose clearance. It remains to be studied whether microbiota depletion also affects

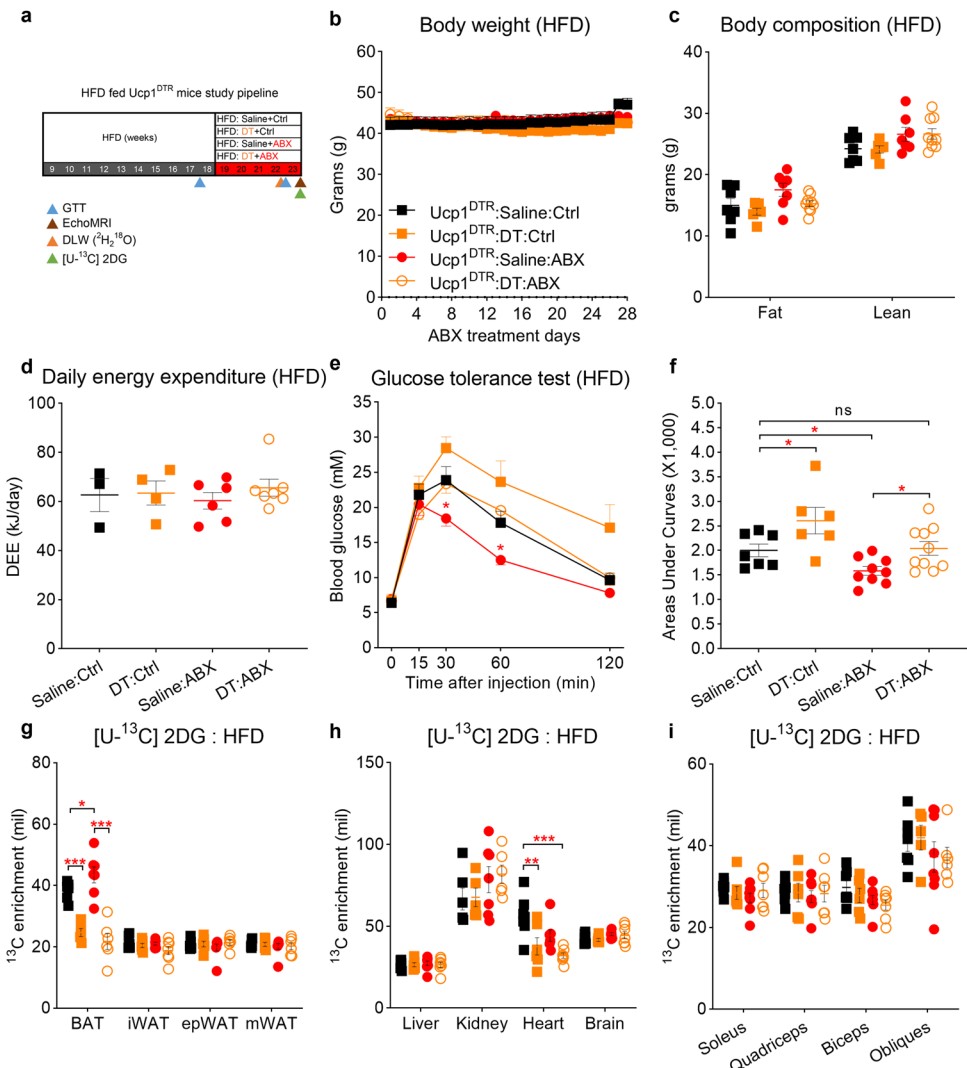

**Fig. 5 Ucp1⁺ cells are indispensable for gut microbiota depletion-induced glucose improvement in obese mice. a** Experimental scheme. 10-weeks HFD-fed Ucp1DTR mice were divided into four groups. Saline-Ctrl group (black square) received subcutaneous saline injections and drank autoclaved water; DT-Ctrl group (orange square) gave subcutaneous DT injection and drank autoclaved water; Saline-ABX group (red circle) received saline injections and ABX treatment; and DT-ABX group (orange open circle) gave subcutaneous DT injection and ABX treatment. The results shown are representative of two independent experiments. **b–d** Body mass curves (**b**), final body composition (**c**), and daily energy expenditure (**d**) of four different treated HFD-fed Ucp1-DTR mice (**b**, **c**: Ucp1DTR:Saline:Ctrl, $n = 7$, Ucp1DTR:DT:Ctrl, $n = 6$; Ucp1DTR:Saline:ABX, $n = 9$; Ucp1DTR:DT:ABX, $n = 9$; **d**: Ucp1DTR:Saline:Ctrl, $n = 3$, Ucp1DTR:DT:Ctrl, $n = 4$; Ucp1DTR:Saline:ABX, $n = 6$; Ucp1DTR:DT:ABX, $n = 7$;). **e**, **f** Intraperitoneal glucose tolerance test (ipGTT) and total glucose area under the curves (AUC) at the 4th week of treatment (Ucp1DTR:Saline:Ctrl, $n = 7$, Ucp1DTR:DT:Ctrl, $n = 6$; Ucp1DTR:Saline:ABX, $n = 10$; Ucp1DTR:DT:ABX, $n = 10$) (**e**: $P = 0.037$ at T = 30 min, $P = 0.045$ at T = 60 min; **f**: Ctrl, $P = 0.012$; DT, $P = 0.012$; Saline, $P = 0.026$;). **g–i** 2DG enrichment in fat depots (**g**), skeletal muscle (**I**), and other tissues (**h**) after intraperitoneal injection of [U-¹³C] labelled 2DG to mice at the 5th week of treatment (Ucp1DTR:Saline:Ctrl, $n = 7$, Ucp1DTR:DT:Ctrl, $n = 6$; Ucp1DTR:Saline:ABX, $n = 8$; Ucp1DTR:DT:ABX, $n = 7$) (BAT:Ctrl, $P < 0.001$; ABX, $P < 0.001$; Saline, $P = 0.027$; Heart:Ctrl, $P = 0.007$;). DT diphtheria toxin, DTR diphtheria toxin receptor. All statistical analyses were performed by two-way ANOVA with Bonferroni's multiple comparisons. All results are given as mean ± SEM. Differences with $P < 0.05$ were considered to be significant. See also Supplementary Fig. 5. $P < 0.05$ (*), $P < 0.01$ (**), and $P < 0.001$ (***).

hepatic gluconeogenesis, since the hepatic glycogen in the pyruvate tolerance test could be fast uptake by BAT and cecum from the circulation in ABX mice.

A recent study proposed that the gut microbiota does not impact adaptive thermogenesis[23] but rather the observed effect is due to an artefact stemming from enlargement of the cecum in ABX-treated individuals. To account for this effect, they evaluated energy expenditure in mice with the cecum removed, or they adjusted total body mass for the intestinal contents measured post-mortem. However, there are several issues with these approaches. In particular cecectomy, enterocyte populations that may be involved in the regulation of energy balance through

communicating with other gut tissues, and production of gut hormones such as secretin[49], glucagon-like peptide 1 (GLP-1)[50]. Moreover, ABX mice and GF mice have impaired absorption efficiency due to the lack of gut microbiota. The extended intestine and enlarged cecum are the physiological compensation to sustain energy assimilation in the microbiota-depleted mice. Therefore, the surgical removal of the cecum might affect the metabolic activity of other tissues and eventually impact on the whole-body energy expenditure. At last, the cecum is also known to be highly metabolically activated when the gut microbiota is depleted[20]. A recent study reported that antibiotic induced microbiota depletion alters gut signaling and cecum metabolism.

They found that gut microbiota depletion promoted a shift of substrate utilization from SCFA to glucose. Consistent with their finding, we also found the cecum of ABX mice imported higher glucose contents than Control mice in both lean and obese state (Fig. 1j and Supplementary Figs. 1i, 3h, 4h).

Adjusting the body mass used in analysis of covariance[51] to account for the cecum contents may be a way around these issues with cecectomy[23]. When we used this approach in the current data, we found that the 'Krisko adjustment' had no impact on the outcome of the analysis (ANCOVA, $F_{1,12} = 0.43$, $P = 0.526$). We previously found that ABX mice had lower total energy expenditure after adjusted with body mass or lean mass using ANCOVA[26]. To evaluate the impact of adjusting the body mass estimates using the 'Krisko adjustment' in this previous experiment, we calculated the adjusted body masses using post-mortem estimates of cecum contents derived from a separate cohort. The previous analysis indicated that there was no significant impact body mass (ANCOVA, $F_{1,23} = 0.49$, $P = 0.49$) but a significant impact of the ABX treatment (ANCOVA, treatment $F_{1,23} = 6.69$, $P = 0.016$). After adjustment of the body masses we also found no effect of mass and the ABX treatment effect became (ANCOVA, $F_{1,23} = 4.19$, $P = 0.052$). Technically this is not statistically significant. However, there are several caveats worth noting before we consider discarding the previous result. First, we used the masses of the cecum contents from a separate cohort hence the applicability of these values to the mice we measured is uncertain. Specifically, we may have overcorrected for the mass of the caecum contents. Second, there is a strong movement away from rigorous application of $P$-value thresholds to establish statistical inference[52–54]. A $P$-value of 0.052 indicates that the value 0 lies just inside the confidence interval of the estimated effect size. Hence, it also means that while these data do not allow us to reject the hypothesis that the effect is 0, they also do not allow us to reject the hypothesis that the effect size is on the upper side of the confidence limit—in this case an effect on the metabolic rate of 14%. If we took a Bayesian approach to the analysis and specified a prior that we expected an effect of ABX on metabolism of 10%, then we would be unable to reject that hypothesis with either the unadjusted or 'Krisko adjusted' data. Finally, since in this experiment there was no significant effect of body mass, it is questionable whether it is actually necessary to include body mass into the ANCOVA model, and if removed the ABX treatment effect becomes highly significant ($F_{1,24} = 18.6$, $P < .001$). We suggest based on these analyses of our previous data, combined with the data presented here that the most parsimonious interpretation is that ABX treatment causes a reduction in metabolic rate, but that this impact disappears when UCP1 is absent.

In our previous study, we found the gut microbiota depletion impaired the Ucp1-dependent thermogenesis by blunting UCP1 expression during acute cold stimulation[26,55]. But UCP1 mediated thermogenesis is not the only mechanism for elevating heat production[7,8], therefore we aimed to evaluate the impact of gut microbiota on UCP1-independent thermogenesis. Acute and chronic (the step-down cooling protocol) cold exposure will activate both Ucp1-dependent and Ucp1-independent thermogenesis[4,5]. We confirmed that ABX treatment impaired the β3-adrenoceptor agonist induced (i.e., UCP1-dependent) adaptive thermogenesis in wild-type mice. Next, we found that microbiota depletion had a negligible impact on UCP1-independent adaptive thermogenesis using Ucp1-KO mice. It is still unclear why the UCP1-independent thermogenic pathway remains intact where UCP1-dependent pathway was dampened in gut microbiota-deficient mice. One possibility is that gut microbiota-derived metabolites were not required for UCP1-independent thermogenesis.

Our data have potential translational implications for treating diabetes. In the last decades, it has been confirmed that adult humans possess BAT depots and the size and activity of BAT is related to ambient temperature and the state of adiposity[12,56,57]. Similar to cold induced thermogenesis, early observations in the 1970s suggested that BAT is involved in diet-induced thermogenesis (DIT) in humans[58] and this prandial thermogenesis has recently been linked to the gut hormone Secretin and plays an important role in regulating satiety[49,59]. Several studies have proposed that physiological or pharmacological activation of BAT thermogenesis leads to elevated energy expenditure, associated with improved glucose metabolism in both rodents and humans[15,56,60]. While it is initially surprising in our study that the beneficial effect of microbiota depletion in glucose metabolism was regulated by BAT, independent of adaptive thermogenesis, there is no a priori reason to assume that the contribution of BAT to the improvement of glucose metabolism should be dependent on activated thermogenesis. Current research efforts have focused on how to activate BAT thermogenesis or recruit more beige/brite adipocytes to improve blood glucose, but some pharmacological agents which increase UCP1 content have side effects such as elevated blood pressure which potentially increases cardiovascular risk[61]. We found that the glucose control function in BAT can be dissociated from the adaptive thermogenesis and UCP1. In this case, our study potentially opens a new avenue for study of the relationship between thermogenesis and glucose metabolism in BAT. Future studies will focus on the glucose regulation mechanism underpinning the gut-BAT axis.

Our study has several limitations that we acknowledge here. First, although we complied over 2700 measurements of $^{13}C$ contents in 22 different tissues using two glucose tracers, we were only able to address that BAT is essential for the gut microbiota mediated glucose improvement. $^{13}C$ glucose metabolic flux was not determined in mice and it is important to understand how BAT utilizes the imported glucose under normal and microbiota-deficient state. It is also possible that a small subset, not all, of brown adipocytes modulated the glucose uptake in ABX mice[62]. Moreover, we only analyzed glucose contents at one-timepoint, which was the relative peak point of $CO_2$ output. It is possible that other tissues are important for uptake and utilization at different time points. Second, it has been documented that, in addition to classic interscapular BAT, more BAT-like and brite/beige adipose depots have been confirmed in mice using an image-guide approach[63]. Using Ucp1-DTR mice, we are able to delete Ucp1+ cells in BAT and these BAT-like depots. It is currently unclear what the ability for glucose uptake is in these BAT-like depots, and is possible that other BAT-like depots also enhanced their glucose uptake capacity when the microbiota was absent. Third, we used $^{13}C$-labeled 2DG as indicator to measure the cumulative glucose uptake. It is important to point out that 2DG disrupts the glucose sensor system and leads to hyperglycemia[64]. It is possible that 2DG contents do not reflect the actual glucose distribution, even though we obtained a similar readout with the $^{13}C$ glucose tracer. Hence, to avoid the impact of 2DG on the glucose metabolism, we perform ipGTT first in HFD-fed Ucp1-KO and Ucp1-DTR mice, then 1 week later mice were injected with $^{13}C$ 2DG tracer, which was mixed with glucose solution. Lastly, all the experiments were performed in C57BL/6 N mice background including Ucp1-KO mice which was originally generated in a mixture of C57 and Sv129 background. Different mouse strains in different house conditions, age, cold exposure programs, antibiotic administration route (drinking water or gavage), diets (LFD or chow), and other factors[65] might have different background in gut microbiota composition leading to different readouts in glucose tolerance tests and energy

expenditure assays. Despite these limitations, our study still explains the role of BAT in gut microbiota depletion mediated glucose improvement and offers some important insights into future studies using ABX mouse models to further investigate the glucose homeostasis mechanisms underpinning the gut-BAT axis.

In summary, our data highlight the important role for gut-BAT axis in regulating glucose metabolism. We confirmed that antibiotic-mediated microbiota depletion improves glucose tolerance in mice. Using isotope labeled glucose tracers, we found that microbiota depletion significantly enhanced the ability of glucose uptake in BAT, rather than in the WAT or liver. Microbiota depletion does not negatively affect UCP1-independent thermogenic mechanisms, unlike the impact on UCP1-dependent processes. Furthermore, this improvement of glucose metabolism in ABX mice does not require UCP1 but can be blocked through selective deletion of brown adipocytes. This suggests substantive UCP1-independent glucose utilization in BAT.

## Methods

**Experimental models**. All animal procedures were approved by the Institute of Genetics and Developmental Biology Chinese Academy of Sciences (IGDB-CAS) Institutional Animal Care and Use Committee (IACUC). All relevant ethical regulations have been complied. All mouse experiments were performed in male mice and all mice were housed in specific-pathogen-free facility (SPF) kept at $23 \pm 1$ °C and 50% humidity with a dark-light cycle of 12 h:12 h (lights on at 07:30) and fed ad libitum with a standard chow diet (20% kcal Protein, 70% kcal Carbohydrate and 10% kcal Fat, #D12450B, Research Diets, New Brunswick, NJ) in all LFD experiments or with a high-fat diet (20% kcal Protein, 20% kcal Carbohydrate, and 60% kcal Fat, #D12492, Research Diets, New Brunswick, NJ) in all HFD experiments. Mice were provided with autoclaved drinking water. Antibiotics were administered in the sterile drinking water ad libitum and changed every 3 days, ABX protocol containing ampicillin (0.5 mg/mL, J&K, Cat# A01-290395), neomycin (0.5 mg/mL, J&K, Cat# A01-557926), gentamicin (0.5 mg/mL, J&K, Cat# A01-405947), metronidazole (0.5 mg/mL, J&K, Cat# J07-M0924), vancomycin (0.25 mg/mL, INALCO, Cat# 1758-9326), and sucralose (4 mg/mL, Splenda, J&K, Cat# A01-442522)[30]. In LFD experiments, antibiotics treatment started at 8–9-weeks-old mice for about 4–5 weeks. In HFD experiments, antibiotics treatment started at 19-week-old obese mice. For acute cold challenge experiment, mice were fed ad libitum and individually housed at 4 °C for 48 h. For the chronic cold challenge experiment, mice were fed ad libitum and individually housed, ambient temperature was decreased from 22 °C to 4 °C (−2 °C per day) which was controlled by a laboratory incubator. In HFD experiments, mice were fed with high-fat diet starting at 9 week old for 10 weeks, then were treated with ABX for about 4–5 weeks.

**Metabolic phenotype analysis**. We used magnetic-resonance whole-body composition analyzer to analyze mice body composition (fat mass, lean mass and water content). The body weight and food intake were measured daily. We assessed energy expenditure and physical activity by using indirect calorimetric system (TSE PhenoMaster, TSE Systems, Bad Homburg, Germany). Daily energy expenditure (DEE) of HFD-fed mice were measured by using the doubly labelled water (DLW, $^2$H$_2$$^{18}$O) technique[37,66]. To assess the effect of b3-adrenergic agonist CL-316243 on the energy expenditure, we treated mice daily for 3 consecutive days through intraperitoneal administration of either vehicle (PBS) or CL-316243 (1 mg/kg, Sigma, Cat# C5976).

**Diphtheria toxin injection**. To delete Ucp1$^+$ cells, Ucp1-DTR mice were subjected to two subcutaneously injection with 400 ng Diphtheria toxin (Sigma, Cat# D0564) or PBS every 8 h at the scapular region for two continuous days. Two days after injection, we repeated the injection route again depending the experimental design.

**Glucose tolerance tests**. Overnight fasting (16 h) mice were received an i.p. injection of 20% glucose solution at dose of 2 g glucose per kg body weight (2 g/kg BW). Blood glucose were taken from tail vein before injection and at 15, 30, 60, and 120 min after injection, then were determined by OneTouch glucometer.

**$^{13}$C enrichment in breath samples**. We have developed a technique to measure the glucose homeostasis by using [U$^{13}$C]–glucose (Cambridge Isotope Lab, Cat# CLM-1396-PK). The mice were fasted overnight before the experiment and transferred to a small chamber with a constant gas flow after the i.p. injection of 2 g/kg BW [U$^{13}$C]–glucose, which were diluted by [U$^{12}$C]–glucose at a ratio of 1:19. The flow is regulated such that the outflow CO$_2$ is around 0.5%. This is 20 × higher than the atmospheric CO$_2$ hence incoming CO$_2$ has only a negligible impact on the

measurements. The outflow stream is then sampled at 10-min intervals by collecting gas into a standard vacutainer for up to 200 min. The air samples were analyzed using an auto-sampler linked to a conventional gas source isotope ratio mass spectrometer (Microgas uG). [U$^{13}$C]–glucose in circulation enters cells and then is metabolized and appears as $^{13}$CO$_2$ in the breath. Hence by collecting breath and measuring the $^{13}$C:$^{12}$C ratio in respiratory CO$_2$, the rate of glucose clearance from the body was measured. This provides real time readout of the glucose in a stress free and more detailed manner than the standard test provides.

**$^{13}$C enrichment in tissue samples**. We developed a isotope-based method allowing us to explore the tissue specific glucose uptake, based on the method used in optical in vivo imaging of glucose metabolism[67]. Glucose uptake into cells can be measured using $^{13}$C universal labelled 2-deoxy-D-glucose ([U-$^{13}$C]–2DG) (Cambridge Isotope Lab, Cat# CLM-10466). 2DG has the hydroxyl group on the second carbon molecule replaced by a hydrogen. It has been known since the 1950s that this change does not impair uptake into cells, via the glucose transporters, but the 2DG cannot be metabolized by the process of glycolysis[32]. The mice were fasted overnight and received an i.p. injection of 2 g/kg BW [U-$^{13}$C]–2DG diluted with [U-$^{12}$C]–glucose at a ratio of 1:9. After 25 min, the mice were anesthetized by 0.1% isoflurane. Then the mice were rapidly dissected within 5 min for the collecting of fresh tissues following a perfusion with sterile saline. The tissues were weighted and dried for 2 weeks in a 60 °C oven. Then whole dry tissues were weighted and ground into powder, which can be analyzed by a mass spectrometer. By measuring the $^{13}$C:$^{12}$C ratio in tissues dry weight, the rate of glucose uptake into different tissues was measured. The [U-$^{13}$C]–glucose was used to measure dynamic glucose homeostasis in different tissues with the same method. To estimate the tissue turnover we used an exponential uptake-elimination model. That is we defined the glucose uptake from the $^{13}$C-2-dG levels, minus the background isotope enrichment, and then calculated turnover from the difference between the log$_e$($^{13}$C-2-dGlucose) and log$_e$($^{13}$C Glucose) at the 30 min timepoint (both expressed minus the background enrichment of $^{13}$C. The scaled C13 heatmaps were generated in R version 4.03.

**Western blotting**. The whole adipose tissue pads were homogenized in tissue protein extraction reagent (ThermoFisher, Cat# 78510) supplemented with a protease inhibitor cocktail (Sigma, Cat# P8340) and 10 mM PMSF (Sigma, Cat# P7626) and the lysates were chilled on ice for 30 min then centrifuged at 13,800 × g (10 min) to remove cell debris. The supernatant was collected then the protein concentration was measured by bicinchoninic acid (BCA) protein quantification kit. Equal amounts of protein were separated on 12% SDS-PAGE gels and blotted onto Immobilon-P PVDF membranes. After blocking in 5% skim milk in PBS-T (PBS with 0.05% Tween-20) for 1 h at room temperature, membranes were incubated with primary antibody at 4 °C overnight. To visualize the bands, HRP-labeled secondary antibodies (Goat anti-Mouse IgG HRP antibody, Cat# ZB-2305, 1:5,000 diluted; Goat anti-Rabbit IgG HRP antibody, Cat#ZB-2301, 1:5,000 diluted) and ECL blotting reagents (GE Healthcare) were used. The bands were quantified with ImageJ 1.50i software. The primary antibodies used for western blot were UCP1 (Abcam, Cat# ab10983, 1:3,000 diluted), β-actin (ZSGB-Bio, Cat# TA-09, 1:5,000 diluted).

**Quantification and statistical analysis**. Replicate information is indicated in the figure legends. All results are given as mean ± SEM and analyzed by using statistical tools implemented in Prism (GraphPad version 8). Statistical analyses were performed using the Student's $t$-test and regular one-way or two-way analysis of variance (ANOVA) with Bonferroni correction. Differences with $P < 0.05$ were considered to be significant. $P < 0.05$ (*), $P < 0.01$ (**), and $P < 0.001$ (***).

**Reporting summary**. Further information on research design is available in the Nature Research Reporting Summary linked to this article.

## Data availability

All raw data generated or analyzed during this study are in this published articles and its supplementary information files. Source data are provided with this paper.

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

## Acknowledgements

This work was supported by the National Natural Science Foundation of China (92057206), the KC Wong Education Foundation, as well as grants from the '1000 talents' recruitment program, a PIFI professorial fellowship from CAS and a Wolfson merit professorship from the UK Royal Society, all to J.R.S. We are grateful to all the members of the Molecular Energetics Group for their support and discussion of the results. We would like to thank Peter Thomson and Marina Stamatiou for technical assistance with the DLW measurements.

## Author contributions

Conceptualization, J.R.S., B.G.L., and L.L.; Methodology, J.R.S., B.G.L., L.L., M.L., and C.H.; Investigation, M.L., L.L., B.G.L., Y.G.W., Z.G.J., A.Y.Q.W., C.Q.N., G.L.W., and C. H., Writing—Original Draft, L.L., M.L., B.G.L., and J.R.S.; Writing, L.L., M.L., B.G.L., C. W., and J.R.S.; Supervision, J.R.S.

## Competing interests

The authors declare no competing interests.
