## [Peer Review File · Nature Communications]

Reviewers' Comments:

Reviewer #1:

Remarks to the Author:

Adaptive thermogenesis is highly dependent on uncoupling protein 1 (UCP1), a protein expressed by thermogenic adipocytes in brown adipose tissue (BAT). It has been shown that adrenergic stimulation can increase glucose uptake in BAT independently of the thermogenic function of UCP1. In addition, it is known that both germ-free and antibiotic-treated (ABX) mice have lower blood glucose levels and higher BAT activity. However, it is not entirely clear whether other organs such as liver, muscle, white adipose tissue, heart or other organs contribute to the glucose-lowering effect. The current manuscript investigates whether the microbiome-dependent effects on plasma glucose levels can be explained primarily by BAT-specific glucose processing. In addition, by using UCP1-deficient mice as well as brown adipocyte-depleted mice, the authors investigated whether microbiome-dependent glucose lowering is associated with UCP1-dependent thermogenesis and/or BAT abundance. Through a series of highly sophisticated approaches, including metabolic flux studies of ¹³C-glucose and ¹³C-deoxyglucose in mice exposed to warm and cold housing temperatures, they show that ABX-mediated depletion of the gut microbiome triggers glucose-dependent processing in BAT. Studies in transgenic mice suggest that this effect is separated from UCP1-dependent thermogenesis. Although the results are somewhat descriptive in nature and the effect sizes is rather small, the overall message of the paper is clear and well described. On the other hand, there are a number of open questions and, more importantly, some of the conclusions drawn are not justified by the data presented.

1. Fig. 1A. Glucose tolerance is shown only for mice housed at 22°C. However, in Fig. 1C-1J, the authors identify the main differences in glucose tracer processing in mice exposed at 4°C. To conclude that the effect of ABX on plasma glucose lowering is mediated by BAT, the authors need to perform glucose tolerance assays in which they show an effect of ABX on glucose lowering at 4°C.
2. Supplemental Figure 1A shows higher glucose accumulation in the liver of ABX-treated mice in the cold. Similarly, significant differences in tracer accumulation are also observed in other tissues such as the heart and gastrointestinal tract. Thus, the conclusion that organs other than BAT are not important sites for glucose disposal is not justified by the data.
- 3) How can the authors rule out that hepatic glucose production is not modulated by ABX and/or ambient temperature, thereby affecting plasma glucose levels? The tracer studies do not rule out such an effect.
4. Figure 4F. Although not significant (probably because of the higher standard deviation), ABX treatment results in a similar reduction in both control and BAT-ablated mice. Thus, the claim that BAT mediates the glucose-lowering effect in response to depletion of the gut microbiota is not justified by the data.
5. Supplemental Figure 4. Data on glucose disposal in gastrointestinal organs are not shown, but based on the organ weights (Supplemental Figure 4D), it cannot be ruled out that in this experiment the lowering of plasma glucose by ABX was mainly caused by intestinal or cecal uptake.

Reviewer #2:

Remarks to the Author:

In the manuscript "Brown Adipose Tissue, but Not White Adipose Tissue or Liver, Regulates Glucose Homeostasis in Microbiota Depleted Mice" by Li et al intended for publication as an original research article in Nature Communications the authors investigate the role of the gut microbiota and UCP1 in mouse glucose homeostasis.

First, glucose uptake in various tissues is studied using glucose labelled with ¹³C in control mice and in mice treated with abx at room temperature and at 4C. In a second set of experiment the influence of the gut microbiota on glucose and energy metabolism in wild-type and UCP1 KO mice fed low fat diet as well as the role of the gut microbiota during cold adjustment in UCP1 KO mice are studied. In a third experiment the influence of interaction between UCP1 and the gut microbiota on glucose and energy metabolism in a high fat diet induced obesity model is studied. In the fourth and final part of the manuscript the authors study the influencer of the gut

microbiota on glucose and energy metabolism in high fat diet fed mice where a diphtheria toxin receptor is expressed under the control of the UCP1 receptor killing cells expressing UCP1 after injection of diphtheria toxin.

In general, the experiments performed seem to be conducted in a good way and the results as well as statistical analyses appear reliable. In my opinion the manuscript contains a lot of interesting data. However, I have several major concerns about the conclusions made, the structure of the manuscript, the way data is presented and the interpretation of the results.

- The statements made in the title are not supported by the presented data. First, regulation and uptake are not the same thing. Regulation can be complex and involve signaling and other types of interaction between cells and organs and is not restricted to flux of glucose. Also, in addition to BAT there are major differences in glucose uptake sections of the intestinal tract (eg caecum). Also, Figure S1A show a small increase in U13C glucose in the liver in abx mice at 4C. Given the larger size of the liver compared to BAT this should be taken into account and the role of the liver in glucose uptake should not be neglected.
- The difference in glucose tolerance between control mice and abx mice at room temperature presented in Figure 1a is not explained by the data for glucose respiration and uptake presented in Fig1. At room temp only minor differences in glucose uptake is shown. BAT show no differences in 13C and only a minor shift in 13-2DG at 22C. At 22C it appears as parts of the intestinal tract differ the most in glucose uptake. Hence, the difference in glucose tolerance at room temperature needs to be explained.
- The GTT (Figure 1A) should also include mice exposed for an acute cold exposure similar to the rest of Figure 1.
- The rationale for using CL316246 to induce thermogenesis in Figure 2 is not properly explained. In Figure 1 cold exposure is used for acute activation of thermogenesis. Why is this model changed?
- For long term cold adaptation wild-type mice should be included to compare UCP1 dependent and independent regulation (Fig2). Why is only the KO presented here?
- Are data in Figure 2K from TSE cages? There is no experiment with metabolic cages indicated in figure 2I.
- Why is only 13C DG monitored in the obesity model while both 13C Glucose and 13C DG are presented in Figure 1? Several interesting difference such as those in the gastro-intestinal tract was seen in glucose (Fig S1B) but not in DG (Fig S1I).
- How is the UCP1-DTR model validated? Western blot (similar to Figure S2a) should be presented at least for BAT. Also, shouldn't amelioration of UCP1 expressing cells decrease the weight of BAT (Figure S4B)?
- How is the ABX model justified? How do the authors know that the antibiotic treatment per se has side effects that affect metabolism independent on the gut microbiota? Critical experiments should be repeated in eg a GF model.
- What diet is used in experiments presented in Figure 1? This is not clearly stated. Is it the same LFD as in Figure 2?
- It is sometimes difficult to understand why experiments have been performed and how the different figures/experiments are related.

Reply to reviewers' comments:

Reviewer #1 (Remarks to the Author):

Comment: Adaptive thermogenesis is highly dependent on uncoupling protein 1 (UCP1), a protein expressed by thermogenic adipocytes in brown adipose tissue (BAT). It has been shown that adrenergic stimulation can increase glucose uptake in BAT independently of the thermogenic function of UCP1. In addition, it is known that both germ-free and antibiotic-treated (ABX) mice have lower blood glucose levels and higher BAT activity. However, it is not entirely clear whether other organs such as liver, muscle, white adipose tissue, heart or other organs contribute to the glucose-lowering effect. The current manuscript investigates whether the microbiome-dependent effects on plasma glucose levels can be explained primarily by BAT-specific glucose processing. In addition, by using UCP1-deficient mice as well as brown adipocyte-depleted mice, the authors investigated whether microbiome-dependent glucose lowering is associated with UCP1-dependent thermogenesis and/or BAT abundance. Through a series of highly sophisticated approaches, including metabolic flux studies of ¹³C-glucose and ¹³C-deoxyglucose in mice exposed to warm and cold housing temperatures, they show that ABX-mediated depletion of the gut microbiome triggers glucose-dependent processing in BAT. Studies in transgenic mice suggest that this effect is separated from UCP1-dependent thermogenesis. Although the results are somewhat descriptive in nature and the effect sizes is rather small, the overall message of the paper is clear and well described.

Response: we appreciate this overall positive evaluation of the work and recognition of the sophistication of the approaches we have used that are not generally available or employed by other laboratories.

For clarification, we would like to point out our novel findings compared to other previous studies.

It has been reported that the gut microbiota depletion improves glucose tolerance. Although the researchers in the field invested substantial effort to study the molecular mechanism behind this phenotype, it remains largely unknown which tissue(s) are essential for improved glucose uptake. Previous studies only looked at one or few tissues, and several important tissues were overlooked especially BAT and muscle. It seems obvious that if you don't look at a particular tissue you may have missed that it is an important contributor. Yet previous work has ascribed the importance of different tissues without consideration of the potential importance of things they didn't look at. Moreover, these data from different studies makes it difficult to estimate the overall contribution of each tissues in glucose uptake and clearance. In contrast to these previous studies, we evaluated glucose uptake in 22 tissues/organs and found BAT & cecum are the key sites for glucose uptake in microbiota depleted mice. It is important to note that we did not exclude the contribution of other important organs such as muscle or liver in glucose uptake/clearance at the normal state, we only concluded BAT and cecum increased its capacity for glucose uptake in the ABX condition. The glucose uptake capacity in other tissues remained insignificant changes at ABX state. Although the changes in BAT and cecum were not enormous, it is worthy of mention that the changes were observed in merely 30min. We found that BAT presented the highest glucose turnover rate across all the tissues (**Fig. 1j**), the long-term contribution of BAT could be underestimated in the 30min assessment.

Although previous studies proposed that WAT depots (*Suarez-Zamorano et al., 2015, Nature Medicine*) or liver (*Krisko et al., 2020, Cell Metabolism*) are the key sites for the improved glucose clearance in ABX mice, these data were only assessed in the wild-type mice and the essential validation experiments to support these notions were absent. Of note, cecum removal only partially blunted the effect of ABX on glucose clearance. We found that BAT and cecum significantly increased glucose uptake in ABX mice. Moreover, we investigated the role of UCP1 protein and Ucp1⁺ cells respectively in glucose uptake and clearance when microbiota was deficient using UCP1-KO and UCP1-DTR mice. Most importantly, we

showed that depletion of Ucp1⁺ cells with DT fully reversed ABX mediated glucose clearance in both lean and obese mice. In our opinion, these findings are more than “associations” and take the data well beyond ‘descriptive’.

We believe our findings have broad implication based on the suggestions that the important role of BAT-cecum-microbiota in regulation of glucose metabolism and they may guide the discovery of the novel thermogenesis-independent pathways in regulating glucose uptake. We thank the reviewer for their constructive criticism, which has already resulted in the addition of extensive new data and discussions in the revised manuscript. We believe these changes have significantly strengthened our work. Meanwhile, we will continue to improve the limitations raised by reviewers via developing new animal models and methods in the future studies.

Comment: On the other hand, there are a number of open questions and, more importantly, some of the conclusions drawn are not justified by the data presented. 1. Fig. 1A. Glucose tolerance is shown only for mice housed at 22°C. However, in Fig. 1C-1J, the authors identify the main differences in glucose tracer processing in mice exposed at 4°C. To conclude that the effect of ABX on plasma glucose lowering is mediated by BAT, the authors need to perform glucose tolerance assays in which they show an effect of ABX on glucose lowering at 4°C.

Response: We have now added the 4°C ipGTT data at **Fig. 1a**. At 4°C, ABX mice also present a faster glucose clearance compare to control mice. The adaptive thermogenesis machinery (**Fig. 1a**) and the expression of uncoupling protein 1 (UCP1) (**Fig. 2c, 3e**) were dispensable for the increased glucose uptake and clearance.

Comment: 2. Supplemental Figure 1A shows higher glucose accumulation in the liver of ABX-treated mice in the cold. Similarly, significant differences in tracer accumulation are also observed in other tissues such as the heart and gastrointestinal tract. Thus, the conclusion that organs other than BAT are not important sites for glucose disposal is not justified by the data.

Response: We apologize for the confusion and have revised the conclusion in the manuscript.

The liver data in **Supplementary Fig. 1a** and **1h** were carried out in ¹³C glucose and ¹³C 2DG respectively. Although there is a small increase in the ¹³C-contents in the liver at 4°C using U-¹³C glucose, we did not detect this result using U-¹³C 2DG (**Supplementary Fig. 1h**). Since ¹³C-2DG data only reflects accumulative glucose uptake, ¹³C-glucose data reflects a combination of glucose uptake and utilization. It is also noteworthy that the liver mass in ABX: 4°C group was decreased compare to Ctrl: 4°C group (**Supplementary Fig. 1e**). Combined these data, we concluded that the glucose uptake in liver was not affected by ABX treatment.

We have also now added these points in the discussion part of the revised manuscript.

Line 159-161, “These data indicated that BAT, heart, and the alimentary tract, rather than WAT depots or the liver, are the most important sites for enhanced glucose disposal when gut microbiota was absent.”

Line 371-381, “Importantly, ABX mice not only had higher glucose uptake in BAT, but also showed elevated glucose uptake in the digestive organs, suggesting these tissues are important for glucose uptake. Deletion of Ucp1⁺ cells fully blocked the beneficial effect of ABX in GTT indicating that BAT is required for the improved glucose clearance when the gut microbiota was absent. We did not determine whether the cecum is also necessary for this action in this study. Although a recent study addressed ABX treatment mice with the cecum removed, cecum removal only partially blunted the effect of ABX on glucose clearance(Krisko et al., 2020). The liver is an important organ for the glycogen synthesis. In current study, we mainly focused on glucose uptake

and glucose clearance. It remains to be studied whether microbiota depletion also affects hepatic gluconeogenesis, since the hepatic glycogen in pyruvate tolerance test could be fast uptake by BAT and cecum from the circulation in ABX mice.”

Comment: 3 How can the authors rule out that hepatic glucose production is not modulated by ABX and/or ambient temperature, thereby affecting plasma glucose levels? The tracer studies do not rule out such an effect.

Response: This is an important point. We and many other investigators have found the main effect in ABX treatment was the improved glucose clearance in the ipGTT. Accordingly, we chose to investigate the glucose clearance and uptake in the current project.

Since BAT and digestive organs are important sites for glucose uptake in ABX mice. A simple ipPTT carried out by Krisko et al can not determine whether hepatic gluconeogenesis was truly affected by ABX treatment. The correct way to address this would be via a glucose/pyruvate clamp experiment. While we agree that the microbiota may also have impacts on the hepatic gluconeogenesis, this question falls beyond the scope of the current paper. We added discussion of this limitation.

Comment: 4. Figure 4F. Although not significant (probably because of the higher standard deviation), ABX treatment results in a similar reduction in both control and BAT-ablated mice. Thus, the claim that BAT mediates the glucose-lowering effect in response to depletion of the gut microbiota is not justified by the data.

Response: For the mean \pm SD values in **Fig. 4f** (now **Fig. 5f**). Saline:Ctrl group was 2.003 ± 0.339 , DT:Ctrl group was 2.608 ± 0.655 , Saline:ABX group was 1.568 ± 0.245 and DT:ABX group was 2.040 ± 0.441 . The statistical analysis was performed via one-way ANOVA ($F_{3,28} = 1.413$). Compare to Saline:Ctrl group (black square), ABX treatment (Saline:ABX group, red circle) significantly lower the AUC in GTT ($p=0.026$) whereas the deletion of Ucp1 cells (DT:ABX group, orange open circle) fully blocked the ABX effect in the glucose clearance ($p=0.862$). The effect can also be witness in **Fig. 5e**.

We noticed that DT:Ctrl group had deteriorated GTT performance, we think the high fat diet induced obesity could be a confounding factor to the GTT performance when UCP1 cells were absent. Hence, we carried out a new experiment by using the lean Ucp1^{DTR} mice. The new data is in **Fig. 4c-d**. In low fat diet fed mice, only the Saline:ABX group showed the improvement of the glucose clearance.

Comment: 5. Supplemental Figure 4. Data on glucose disposal in gastrointestinal organs are not shown, but based on the organ weights (Supplemental Figure 4D), it cannot be ruled out that in this experiment the lowering of plasma glucose by ABX was mainly caused by intestinal or cecal uptake.

Response: For the ¹³C 2DG in the digestive organs, the data was shown in **Supplementary Fig. 4h** (now **Supplementary Fig. 5h**) and mentioned at **line 305-308**. Specifically <Consistent with the ¹³C 2DG result in ABX treated obese wildtype mice at **Supplementary Fig. 3h**, the Saline-ABX group (2-way ANOVA, $F_{3,24} = 6.82$, $P < 0.001$) and DT-ABX group (2-way ANOVA, $F_{3,24} = 6.82$, $P < 0.001$) both had increased ¹³C 2DG contents in the cecum compared to the 2 non-ABX treated groups (**Supplementary Fig. 5h**).>

Together, the data in **Supplementary Fig. 5h** and **Fig. 5e/5f** suggested the digestive organs are important sites for glucose uptake in ABX mice, but they are not the key sites for modulating glucose clearance in ABX treatment.

Reviewer #2 (Remarks to the Author):

Comment: In the manuscript “Brown Adipose Tissue, but Not White Adipose Tissue or Liver, Regulates Glucose Homeostasis in Microbiota Depleted Mice” by Li et al intended for publication as an original

research article in Nature Communications the authors investigate the role of the gut microbiota and UCP1 in mouse glucose homeostasis. First, glucose uptake in various tissues is studied using glucose labelled with ^{13}C in control mice and in mice treated with abx at room temperature and at 4°C . In a second set of experiment the influence of the gut microbiota on glucose and energy metabolism in wild-type and UCP1 KO mice fed low fat diet as well as the role of the gut microbiota during cold adjustment in UCP1 KO mice are studied. In a third experiment the influence of interaction between UCP1 and the gut microbiota on glucose and energy metabolism in a high fat diet induced obesity model is studied. In the fourth and final part of the manuscript the authors study the influence of the gut microbiota on glucose and energy metabolism in high fat diet fed mice where a diphtheria toxin receptor is expressed under the control of the UCP1 receptor killing cells expressing UCP1 after injection of diphtheria toxin.

In general, the experiments performed seem to be conducted in a good way and the results as well as statistical analyses appear reliable. In my opinion the manuscript contains a lot of interesting data.

Response: we are grateful for this overall positive evaluation of our studies.

Comment: However, I have several major concerns about the conclusions made, the structure of the manuscript, the way data is presented and the interpretation of the results.

1)- The statements made in the title are not supported by the presented data. First, regulation and uptake are not the same thing. Regulation can be complex and involve signaling and other types of interaction between cells and organs and is not restricted to flux of glucose. Also, in addition to BAT there are major differences in glucose uptake sections of the intestinal tract (eg caecum). Also, Figure S1A show a small increase in ^{13}C glucose in the liver in abx mice at 4°C . Given the larger size of the liver compared to BAT this should be taken into account and the role of the liver in glucose uptake should not be neglected.

Response: We agree with these comments and have revised the title and conclusion accordingly. It now reads “Brown Adipose Tissue, but Not White Adipose Tissue, is the Key Depot for Glucose Clearance in Microbiota Depleted Mice”.

Although there is a small increase in the ^{13}C -contents in the liver at 4°C using ^{13}C glucose, we did not detect this result using ^{13}C 2DG (**Supplementary Fig. 1h**). Since ^{13}C -2DG data only reflects glucose uptake, ^{13}C -glucose data reflects a combination of glucose uptake and utilization. It is also noteworthy that the liver mass in ABX: 4°C group was decreased compare to Ctrl: 4°C group (**Supplementary Fig. 1e**). Combined these data, we concluded that the glucose uptake in liver was not affected by ABX treatment.

Comment: 2)- The difference in glucose tolerance between control mice and ABX mice at room temperature presented in Figure 1a is not explained by the data for glucose respiration and uptake presented in Fig 1. At room temp only minor differences in glucose uptake is shown. BAT show no differences in ^{13}C and only a minor shift in ^{13}C -2DG at 22°C . At 22°C it appears as parts of the intestinal tract differ the most in glucose uptake. Hence, the difference in glucose tolerance at room temperature needs to be explained.

Response: Because only ^{13}C -2DG contents in BAT and the digestive organs were increased at room temperature this demonstrated both sites are important for the glucose uptake. We agree that the experiments in **Fig 1** are unable to determine which of the high glucose uptake sites is responsible to the lower blood glucose in ipGTT (Reviewer 1 point 2).

We added some extra discussion to address this concern at **line 371-381**.

Line 371-381, “Importantly, ABX mice not only had higher glucose uptake in BAT, but also showed elevated glucose uptake in the digestive organs, suggesting these tissues are important for glucose uptake. Deletion of Ucp1^+ cells fully blocked the beneficial effect of ABX in GTT indicating that BAT is required for the improved glucose clearance when the gut microbiota was absent. We did not determine whether the cecum is also necessary for this action in this study.

Although a recent study addressed ABX treatment mice with the cecum removed, cecum removal only partially blunted the effect of ABX on glucose clearance (Krisko et al., 2020). The liver is an important organ for the glycogen synthesis. In current study, we mainly focused on glucose uptake and glucose clearance. It remains to be studied whether microbiota depletion also affects hepatic gluconeogenesis, since the hepatic glycogen in pyruvate tolerance test could be fast uptake by BAT and cecum from the circulation in ABX mice.”

Comment: 3)- The GTT (Figure 1A) should also include mice exposed for an acute cold exposure similar to the rest of Figure 1.

Response: We have added the 4°C ipGTT data at **Fig. 1a**. At 4°C, ABX mice also present a faster glucose clearance compare to control mice. This reduction after ABX treatment was independent of the adaptive thermogenesis (**Fig. 1a**) and UCP1 protein (**Fig. 2c, 3e**). (Also see reviewer 1, point 1)

Comment: 4)- The rationale for using CL316246 to induce thermogenesis in Figure 2 is not properly explained. In Figure 1 cold exposure is used for acute activation of thermogenesis. Why is this model changed?

Response: We apologize for the confusion. We have expanded the relevant sentence in the manuscript at **line 181-186**.

It now reads “Although we previously reported that the gut microbiota is required for Ucp1-dependent thermogenesis of BAT in wild type mice (Li et al., 2019), a recent study has challenged this notion and reported the gut microbiota is dispensable to β 3-adrenoceptor activation in the commensal-depleted (CD) mice (Krisko et al., 2020). Despite the difference in the experimental animals, one hour stimulation of CL-316243 is insufficient to evaluate the activation of thermogenesis. To address these conflicting results, we.....”

Comment: 5)- For long term cold adaptation wild-type mice should be included to compare UCP1 dependent and independent regulation (Fig2). Why is only the KO presented here?

Response: We agree it is important to evaluate UCP1 dependent and independent mechanism on energy expenditure in ABX treated mice. We have previously published the cold stimulated ABX experiment using wild-type (Li et al., 2019) but the outcome in Ucp1-KO mice was not determine. Moreover, due to Ucp1-dependent mechanism is the first responder and dominated pathway in wild-type mice, which presented a confounder for determining when Ucp1-independent mechanism was engaged and its subsequent contribution to the whole body energy expenditure. Hence, the experiment design in **Fig. 2i-2l** we investigated the impact of ABX on Ucp1-independent energy expenditure using Ucp1-KO mice only.

Comment: 6)- Are data in Figure 2K from TSE cages? There is no experiment with metabolic cages indicated in figure 2I.

Response: We apologize for this omission. Energy expenditure at 22°C was measured with TSE metabolic chambers and doubly labeled water was used for the step-cooling mice. Although two methods were used, we are not comparing the data across different temperatures.

Comment: 7)- Why is only ^{13}C DG monitored in the obesity model while both ^{13}C Glucose and ^{13}C DG are presented in Figure 1? Several interesting difference such as those in the gastro-intestinal tract was seen in glucose (Fig S1B) but not in DG (Fig S1I).

Response: Since 2DG can only be taken up and not utilized by the cells or re-enter the circulation, we chose ^{13}C -2DG to focus on the impact of ABX on the glucose uptake.

Comment: 8)- How is the UCP1-DTR model validated? Western blot (similar to Figure S2a) should be presented at least for BAT. Also, shouldn't amelioration of UCP1 expressing cells decrease the weight of BAT (Figure S4B)?

Response: We have added the western blot data for Ucp1-DTR model in **Supplementary Fig. 4a**.

We also noticed that the weight of BAT did not decrease in DT treated obese mice. We think the primary reason is DT only kills the Ucp1 positive cells in BAT. There are several different cell types in BAT to support BAT function such as collagen fibers, vascular cells, immune cells and nerves. HFD treatment will also provide opportunity for white adipocytes to expand in BAT. Hence, we carried out a new experiment by using the lean Ucp1^{DTR} mice. The new data is in **Supplementary Fig. 4d**. Together, we found the deletion of Ucp1 cells lowers BAT weight in the LFD fed mice but not the HFD fed mice. This result was also observed by others (Challa et al., 2020).

Comment: 9)- How is the ABX model justified? How do the authors know that the antibiotic treatment per se has side effects that affect metabolism independent on the gut microbiota? Critical experiments should be repeated in eg a GF model.

Response: To verify the ABX model in our study, we previously assessed the 16S rDNA contents per the fecal pellet in control and ABX mice using qPCR and 16S rDNA amplicons sequencing (Li et al., 2019). The total 16S rDNA contents in feces dropped ~1000 folds from 4.0E+07 to 5.0E+04 (pg DNA per gram feces). The 16S rDNA amplicons sequencing indicated ABX treatment efficiently depleted the majority of gut microbiota.

The key result of the improvement ipGTT in ABX mice has been previously reported in the GF mice (Backhed et al., 2004; Suarez-Zamorano et al., 2015). However, we are limited by the manipulation methods in the GF models. We couldn't perform the cold stimulation or measure the energy expenditure in GF mice because it is impossible to move the GF mice chamber system into a separate cold room or combine it with the metabolic system.

Comment: 10)- What diet is used in experiments presented in Figure 1? This is not clearly stated. Is it the same LFD as in Figure 2?

Response: We apologize for this omission. All mice in figure 1 were fed with LFD. We have clarified this in the manuscript.

Comment: 11)- It is sometimes difficult to understand why experiments have been performed and how the different figures/experiments are related.

Response: We have revised the manuscript with more details. We appreciate the advice and criticism which have strengthened the manuscript.

References

- Backhed, F., Ding, H., Wang, T., Hooper, L.V., Koh, G.Y., Nagy, A., Semenkovich, C.F., and Gordon, J.I. (2004). The gut microbiota as an environmental factor that regulates fat storage. *Proc Natl Acad Sci U S A* *101*, 15718-15723.
- Challa, T.D., Dapito, D.H., Kulenkampff, E., Kiehlmann, E., Moser, C., Straub, L., Sun, W., and Wolfrum, C. (2020). A Genetic Model to Study the Contribution of Brown and Brite Adipocytes to Metabolism. *Cell Rep* *30*, 3424-3433 e3424.
- Krisko, T.I., Nicholls, H.T., Bare, C.J., Holman, C.D., Putzel, G.G., Jansen, R.S., Sun, N., Rhee, K.Y., Banks, A.S., and Cohen, D.E. (2020). Dissociation of Adaptive Thermogenesis from Glucose Homeostasis in Microbiome-Deficient Mice. *Cell Metab* *31*, 592-604 e599.

Li, B., Li, L., Li, M., Lam, S.M., Wang, G., Wu, Y., Zhang, H., Niu, C., Zhang, X., Liu, X., et al. (2019). Microbiota Depletion Impairs Thermogenesis of Brown Adipose Tissue and Browning of White Adipose Tissue. *Cell Rep* 26, 2720-2737 e2725.

Suarez-Zamorano, N., Fabbiano, S., Chevalier, C., Stojanovic, O., Colin, D.J., Stevanovic, A., Veyrat-Durebex, C., Tarallo, V., Rigo, D., Germain, S., et al. (2015). Microbiota depletion promotes browning of white adipose tissue and reduces obesity. *Nature medicine* 21, 1497-1501.

Reviewers' Comments:

Reviewer #1:

Remarks to the Author:

The authors have responded to my criticisms in an appropriate manner. I have no further comments on this interesting paper.

Reviewer #2:

Remarks to the Author:

In the resubmission of the manuscript the authors have addressed most of my concerns about the initial version. Importantly, the claim that brown adipose tissue is the key regulator of glucose homeostasis has been removed. Important clarifications have been made in the manuscript and even though some of the experiments that I was asking for has not been performed the authors have justified why this has not been done.

In my opinion the ms is now suitable for publication.